

# Development of a Curled Wake of a Yawed Wind Turbine under Turbulent and Sheared Inflow

Paul Hulsman[1], Martin Wosnik[2], Vlaho Petrović[1], Michael Hölling[1], and Martin Kühn[1]

[1]ForWind – Institute of Physics, University of Oldenburg, Küpkersweg 70, 26129 Oldenburg, Germany

[2]University of New Hampshire, Department of Mechanical Engineering, S102 Chase Ocean Engineering Laboratory, 24 Colovos Road, Durham NH 03824, United States

**Correspondence:** Paul Hulsman, paul.hulsman@forwind.de

**Abstract.**

A potential technique to reduce the negative wake impact is to redirect it away from a downstream turbine by yawing the upstream turbine. The present research investigated the wake behaviour for three yaw angles $[-30°, 0°, 30°]$ at different inflow turbulence levels and shear profiles under controlled conditions. Experiments were conducted using a model wind turbine with 0.6 m diameter $(D)$ in a wind tunnel. A short-range dual-Doppler Lidar WindScanner facilitated mapping the wake with a high spatial and temporal resolution in vertical, cross-stream planes at different downstream locations and in a horizontal plane at hub height. This versatile equipment enabled the fast measurements at multiple locations in comparison to the well known hot-wire measurements. The flow structures and the energy dissipation rate of the wake were measured from $1\,D$ up to $10\,D$, and for one inflow case up to $16\,D$, downstream of the turbine rotor. A strong dependency of the wake characteristics on both the yaw angle and the inflow conditions was observed. In addition, the curled wake that develops under yaw misalignment due to the counter-rotating vortex pair was more pronounced with a boundary layer (sheared) inflow condition than for uniform inflow with different turbulence levels. Furthermore, the lidar velocity data and the energy dissipation rate compared favourably with hot-wire data from previous experiments with a similar inflow condition and wind turbine model in the same facility, lending credibility to the measurement technique and methodology used here. The measurement campaign provided a deeper understanding of the development of the wake at different inflow conditions, which will advance the process to improve existing wake models.

## 1 Introduction

Wind turbines within a wind farm can experience significant power losses due to wake effects caused by upstream turbines (Barthelmie et al. (2010)) and higher fatigue loads (Thomsen and Sørensen (1999)). Reducing the wake effects of upstream turbines on downstream turbines can potentially increase the power output and decrease the fatigue loads. Currently, one promising technique for mitigating the wake effects is to intentionally steer the wake of the upstream turbine away from the downstream turbine by yawing the rotor of the upstream turbine (Fleming et al. (2014)). Recent studies have shown





that intentionally steering the wake can increase the Annual Energy Production (AEP), Gebraad et al. (2017), and the power output in the free-field, Fleming et al. (2019).

However, developing a yaw-control model to change the trajectory of the wake is a challenging task and requires a thorough understanding of the behaviour of the wake under these conditions. Vollmer et al. (2016) concluded from large-eddy simula-
tions that the interrelation between the atmospheric stability and the yaw misalignment have a strong influence on the wake characteristics, thus affecting the wake deficit, trajectory and wake profile.

Experimental data under controlled conditions is necessary to understand the wake behaviour at different yaw angles and at different inflow conditions, which is critical for developing a steady and reliable yaw control model easy to implement in the field. The potential to redirect the wake has been shown in several wind tunnel experiments, Medici and Alfredsson
(2006), Bartl et al. (2018) and Schottler et al. (2018). One of the earlier studies conducted by Grant et al. (1997) tracked the tip vortices using optical methods downstream of a model turbine. The study indicated the influence of the yaw angle on the tip vortices and thus on the wake expansion and wake deflection. Grant and Parkin (2000) used phase-locked particle image velocimetry (PIV) measurements to measure the circulation within the wake. The measurements found an asymmetry in the wake shape for positive and negative yaw angles. Similar results were obtained by Haans et al. (2005). Medici and Alfredsson
(2006) quantified the velocity deficit at multiple downwind locations in the far-wake region using a two-component hot wire. During their measurements, they observed additional vortex shedding at large yaw angles. Furthermore, they determined that a cross-stream flow component causes the wake to deflect. The PIV measurement campaign conducted by Bastankhah and Porté-Agel (2016) showed an asymmetric flow entrainment in the wake with regard to the mean and turbulent momentum balances. Schottler et al. (2016) investigated the interaction between two model turbines showing clear asymmetries of the
power output of the downstream turbine between a positive or negative yaw angle of the upstream turbine.

In addition, Lundquist and Bariteau (2015) highlighted the importance of measuring the dissipation rate within a wake for modelling purposes, indicating a higher dissipation rate within the wake in comparison to the flow upwind of the turbine. This has also been shown by Neunaber (2019), who used hot-wires to measure the turbine wake.

More recently, studies were conducted estimating the wake deflection downstream for different turbulence and shear con-
ditions by Bartl et al. (2018), who used laser Doppler anemometry to measure the wake characteristics of a yawed turbine below rated wind speed. The three-dimensional flow was investigated in two planes at a downstream distance of three and six times the rotor diameter ($D$) while varying yaw angles between $\psi = -30°, 0°, 30°$. A counter-rotating vortex pair (CVP) was detected for all inflow conditions at large yaw angles. In addition, the measurements indicated that the curled wake shape and wake deflection was more pronounced at low turbulence compared to high turbulence inflow conditions. Furthermore, a
CVP was also detected by Schottler et al. (2018) in a study that compared the wake shapes of two different model wind turbines.

The goal of this paper is to provide a further understanding of the evolution of the curled wake in order to improve wake steering algorithms. The objective is to determine the effect of the boundary layer and turbulence intensity at different yaw angles on the wake deficit, wake deflection and wake dissipation. The inflow conditions were varied by having no grid, a
passive uniform grid, and a passive variable open area grid installed at the test section entrance. The assessment of the wake





was conducted by investigating the mean longitudinal flow component and the turbulent kinetic energy dissipation rate. The mean quantities were used to analyze the wake deficit and wake deflection. The dissipation rate was used to evaluate the evolution and decay of the wake.

Hulsman et al. (2020) reported on the measurement procedure of the Lidar WindScanner, the layout of the wind tunnel and
initial results for one inflow condition. This paper focuses on the the wake behaviour for different inflow conditions. It first describes the setup of the measurement campaign, describing the equipment, the scanning trajectory of the WindScanner and the methodology used for the data acquisition in Section 2. This is followed by the analysis and discussion of the results in Section 3, describing the propagation of the flow through the wind tunnel and the curled wake characteristics for different inflow conditions and yaw angles. A validation and uncertainty analysis of the un-deflected wake measurements is performed
by comparison to literature in Section 4, prior to the conclusions.

## 2 Methodology

The setup of the wind tunnel, the WindScanner (including the scanning trajectories) and the hot-wires for this measurement campaign is described in Section 2.1. An overview of the different measurement cases is shown in Section 2.2. The procedure to determine the wake centre is described in Section 2.3 and the description of how the turbulent kinetic energy dissipation rate
was estimated from the data is given in Section 2.4

### 2.1 Measurement Setup

The measurement campaign was conducted in the large wind tunnel at ForWind-University of Oldenburg (see Figure 1) The wind tunnel has a test section cross-section with the dimensions of $3\,\mathrm{m}\,(H)$ x $3\,\mathrm{m}\,(W)$. For this study three movable test section elements of 6 m length were attached for a total enclosed length of 18 m. The roof of the test section was adjusted to compensate
for boundary layer growth to achieve a zero pressure gradient for the target wind speed of the experiments, nominally 7.5 m/s, with an empty tunnel with no grid or turbine installed. The three-bladed MoWiTO 0.6 wind turbine model (Schottler et al. (2016)), with a hub height $(h)$ of 0.77 m and $0.58D$ m was placed at a distance of $2.4\,D$ downstream of the test section inlet, where the distance was measured to the centre of the rotor. The flow blockage, based on rotor swept area and tower flow-facing area, was 2.7%. The wind turbine controller is based on the torque of the generator (Petrović et al. (2018)) leading to a tip
speed ratio of 5.7 at the operational point. The WindScanner was placed on a robust steel platform about 9 m downstream of the 18 m long test section, near the exit nozzle leading to the wind tunnel return leg.

Three inflow conditions were generated during this campaign with different turbulence levels and shear exponents, by either having no grid, a passive grid with uniform open area, or a passive variable open area grid installed at the downstream end of the wind tunnel nozzle/inlet to the test section. The setup is similar to the study of Neunaber (2019), in order to have a
direct comparison to wake data (for the same turbine design) acquired with a more conventional wind tunnel flow measurement technique (hot wires, constant temperature anemometry). A more detailed description of the inflow conditions and the



measurement cases will be given in Section 2.2.

The WindScanner is a continuous-wave coherent Doppler Lidar and measures a projected line-of-sight component ($v_{\mathrm{LOS}}$) at a sampling rate of 451,7 Hz. The WindScanner can measure between approximately 10 m to 30 m, with a measurement

volume length of 3,1 cm at 10 m focus distance, as shown by van Dooren et al. (2017). It allows measurements of airflow velocity at reasonably high temporal and spatial resolution without disturbing the flow. Two glass prisms which are rotated by a motor to direct the laser beam to every point in space within a cone with an opening angle of $120°$. The WindScanner measures along-beam velocity. Equations 2 approximates the streamwise velocity component $u$ from the measurements, using the assumption that the lateral ($v \approx 0$) and vertical velocity component ($w \approx 0$) are negligible. This assumption fits better the

further downstream the measurement point is located. Here $\gamma$ is the beam elevation angle and $\theta$ is the beam azimuth angle. The line-of-sight component is extracted from the raw Doppler spectrum using the centroid method and includes contributions from all three velocity components, see Equation 1. Furthermore, the focus point of the WindScanner was calibrated before the measurements campaign and the measurement locations were verified using infrared sensitive equipment.

$$V_{\mathrm{LOS}} = u\cos(\gamma)\cos(\theta) + v\sin(\theta)\cos(\gamma) + w\sin(\gamma) \tag{1}$$

$$u = \frac{V_{\mathrm{LOS}}}{\cos(\gamma)\cos(\theta)} - \frac{v\sin(\theta)\cos(\gamma)}{\cos(\gamma)\cos(\theta)} - \frac{w\sin(\gamma)}{\cos(\gamma)\cos(\theta)} \approx \frac{V_{LOS}}{\cos(\gamma)\cos(\theta)} \tag{2}$$

To measure the wake characteristics with the WindScanner, multiple vertical scans at six downstream locations and one horizontal scan at hub height downstream of the MoWiTO 0.6 were performed. This creates a three-dimensional presentation of the evolution of the streamwise velocity component of the wake downstream of the model wind turbine. An illustration of the horizontal scan is seen in Figure 1b, which has a trapezoidal shape with a width of $1.5\,D$ at one side and $3\,D$ at the other

side. Furthermore, the dimensions of the vertical plane is shown in Figure 1c with an area equalling to approximately $3\,D$ x $3\,D$ (1.74 m x 1.74 m). These dimensions were selected to ensure that the development of the boundary layer on the ground and the deflection of the wake at positive and negative yaw angles was captured by the WindScanner. For the analysis of the wake characteristics the acquired velocity data was interpolated onto a grid with a spacing of 7 x 7 cm, which can be considered the structural resolution of the results presented here. For the vertical cross-section scan visualized in Figure 1c the duration for

each Lissajous trajectory scan was approx 7-8 s. This was repeated for 10 minutes for a total of 75 to 85 scans resulting to ≈300 points to the centre grid cell at hub height. For the horizontal planar scan presented in Figure 1b the duration for each Lissajous trajectory scan was approx 22 s and was repeated for 30 minutes.

In addition to the measurements with the WindScanner, hot-wire measurements have been conducted to validate the data

from the WindScanner. Measurements have been conducted with a boundary layer inflow condition (Table 1) and a yaw angle of $\psi = 30°$ using $1\,D$ hot-wire anemometers operated by different systems from Dantec Dynamics and using A/D converters from National Instruments. The anemometers, mounted on a traverse structure, were used to measure a vertical plane at $5\,D$



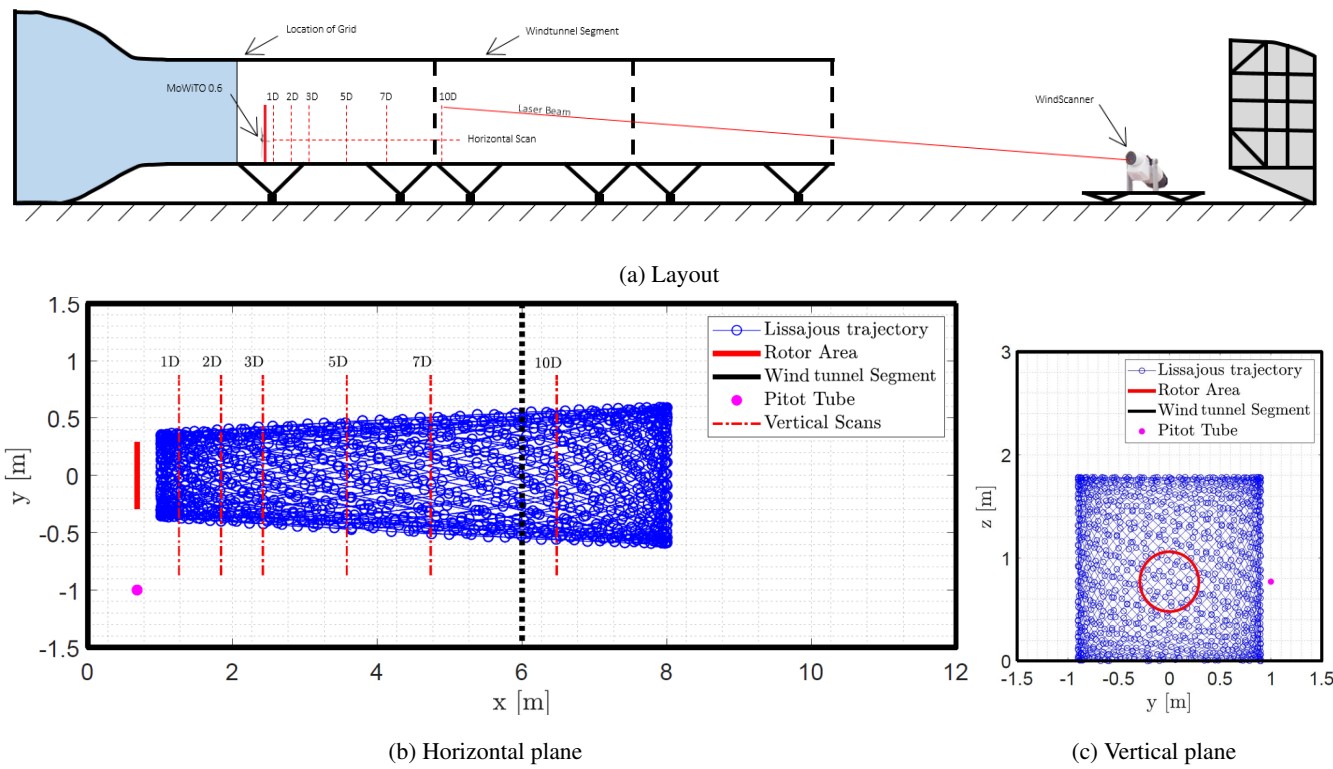

**Figure 1.** The layout of the wind tunnel campaign and the corresponding Lidar scans to measure the flow. A Lissajous trajectory is used to measure a horizontal plane behind the turbine with a duration $t_{\mathrm{scan}} \approx 22$ s per individual scan and a vertical plane with a duration of $t_{\mathrm{scan}} \approx$ 7-8 s per individual scan. The blue dots indicate the individual measurement points along the trajectory of the laser beam.

with a spatial spacing of 3 x 3 cm between $0.25 < z < 1.36$ and $-0.55 < z < 0.55$. The traverse structure was mounted with 20 hot-wires measuring at two spanwise locations, which could be shifted to measure a vertical plane. The measurements at $z = 0.82$ m were not included due to a faulty hot-wire, which lead to a total of 37 x 39 measurements. The hot-wires were measuring at a sampling frequency of 15 kHz for a duration of 120 s. A pitot tube was used to calibrate the hot-wires every

5  three hours during a measurement period. In order to account for the blockage of the setup the pitot tube was located away from the traverse structure during the calibration.

## 2.2   Measurement Cases

Different inflow conditions were achieved by operating the wind tunnel without a grid, a passive grid with uniform open area or a passive grid with variable open area installed at the inlet to the test section. This allowed the yawed wake characteristics and the influence on the CVP to be analyzed at different turbulence levels, uniform inflow and sheared inflow. The passive

10  grid has 100 mm square openings with a spacing of 115 mm and a solidity of 24.4%. The passive grid with variable open area was achieved by installing the ForWind 3m x 3m active grid. The active grid has the possibility to create specific turbulence





patterns and repeatedly impress the flow of the air as described by Heißelmann et al. (2016). In this case the active grid was used in a passive mode, with progressively increasing open area from the floor of the wind tunnel up, to mimic a boundary layer within the segment. For the measurement cases of no grid and the passive grid with uniform open area, the small gaps between the wind tunnel nozzle and the first test section element were sealed with aluminium tape. With the active grid frame

installed there are small gaps that cannot be completely sealed.

Table 1 gives an overview of the different inflow conditions. The test section wind speed was $7.5\frac{m}{s} \pm 0.15\frac{m}{s}$ at hub height for all measurement cases. The test section wind speed was measured with a reference pitot tube at hub height at the rotor plane and 0.5 m from the sidewall, or $1.22D$ to the right of the rotor looking in the streamwise direction. The turbulence intensity ($TI$) and the shear exponent ($\alpha$) at hub height are determined through the WindScanner. This was done in order to obtain the

highest $c_{\mathrm{T}}$ for this model turbine, which leads to the largest wake deflection described in Bastankhah and Porté-Agel (2016). Each inflow condition was measured with and without the model turbine, in order to investigate the propagation of the flow field within the segments. Besides altering the inflow conditions the operational conditions of the turbine model were also modified by changing the yaw angle ($\psi$). The turbine was yawed at $\psi = -30°, 0°, 30°$. The turbine yaw angle is defined as positive when the turbine nacelle is rotated in the clockwise direction when looking at the turbine from above. This leads to

3 x (3+1)=12 unique measurement conditions. For the case with no grid the flow was measured in a horizontal plane at hub height and in vertical planes at $1D$, $2D$, $3D$, $5D$, $13D$ and $16D$, with and without the turbine installed. An additional plane was measured at $0D$ for the cases without a turbine. For the measurement case with the uniform passive grid and the sheared inflow the flow is measured at $1D$, $2D$, $3D$, $5D$, $7D$ and $10D$, with an additional case at $0D$ without a turbine. The measurements were performed plane-by-plane and not simultaneously. In addition to the vertical planes and the horizontal

planes, staring mode measurements at a single point were also performed at $1D$, $2D$, $3D$, $5D$, $7D$ and $10D$ at hub height for a duration of 10 min to obtain data with a frequency of 451,7 Hz.

|  | Variable | $0D$ | $1D$ | $2D$ | $3D$ | $5D$ | $7D$ | $10D$ |
|---|---|---|---|---|---|---|---|---|
| Uniform, No Grid | TI [%] | 0.39 | 0.34 | 0.42 | 0.42 | 0.40 | 0.37 | 0.27 |
| Uniform, Passive Grid | TI [%] | 2.17 | 1.45 | 1.12 | 0.99 | 0.81 | 0.73 | 0.68 |
| Boundary Layer, Passive Grid | TI [%] | - | 1.25 | 1.22 | 1.23 | 1.21 | 1.21 | 1.25 |
| Boundary Layer, Passive Grid | $\alpha$ [-] | 0.28 | 0.29 | 0.29 | 0.29 | 0.28 | 0.29 | 0.28 |

**Table 1.** Overview of the different inflow conditions used during the campaign. The turbulence intensity ($TI$) and the shear exponent ($\alpha$) were obtained from the WindScanner data.

### 2.3 Wake Centre Detection

Two different methods were used to determine the wake centre. The first method determines the wake centre by calculating the position of the minimal potential power of a virtual downstream turbine, described by Schottler et al. (2017). The potential

power in the wind ($P^*$) is determined with Equation 3, which divides the rotor area in five ring segments. Within the area ($A_i$)





of a ring segment, the spatially and temporally averaged velocity $(u_i)_{A_i,t}$ and the air density $(\rho)$ are used to calculate the power. The potential power is the result of the summation of the power determined for each individual ring segment with a width of $6\,cm$, which is below the resolution of the interpolation grid. The location of the lowest potential power (and hence the wake centre) is obtained by computing the potential ring at hub height in the range between $-1\,D < y < 1\,D$.

$$P^* = \sum_{i=1}^{5} \frac{1}{2} \rho A_i (u_i(t))^3_{A_i,t} \tag{3}$$

The second method is performed by extracting the temporally averaged velocity deficit across a horizontal line at hub height from the vertical scan and fitting it with a single peak Gaussian at each downstream distance, similar to the method used by Fleming et al. (2014).

### 2.4  Energy Dissipation Rate Estimation

The turbulent kinetic energy dissipation rate $\varepsilon$ indicates the amount of energy lost due to the viscous forces in a turbulent flow. $\varepsilon$ can determined by using the raw Doppler spectrum width acquired from the WindScanner for each measurement case, in order to analyze the behaviour of the wake for different inflow and turbine operating conditions. The energy dissipation rate was determined using Equation 5 derived by Banakh and Smalikho (1999). Here $\sigma_s$ is the spectrum width, $C \approx 1.5$ is the Kolmogorov constant, $l$ is the Rayleigh length corresponding to the filtering of the small scale turbulence. The Rayleigh length

for a continuous wave Lidar is determined with Equation 4, where $\lambda_b\,(= 1.55\mu\,m)$ is the wavelength of the laser, $r_b\,(= 56mm)$ is the lens aperture radius and $d_f$ is the distance of the measurement point. $\sigma_s$ is determined by fitting a Gaussian distribution over the raw spectrum. The limitation of Equation 5 is that it can only be used when the Rayleigh length is smaller than the large scale turbulence occurring in the flow. For the experiments reported here the Rayleigh length ranges from $l = 94\,mm$ at $x/D = 0$ to $l = 54\,mm$ at $x/D = 10$. Furthermore, with this expression for $\varepsilon$ the mean velocity gradient within the probe

volume is not accounted for.

$$l = \frac{\lambda_b d_f^2}{\pi r_b^2} \tag{4}$$

$$\varepsilon = \left( \frac{\sigma_s^2}{1.22 C l^{2/3}} \right)^{\frac{3}{2}} \tag{5}$$

Hot-wire measurements acquired by Neunaber (2019) in a similar setup and during this campaign were used to determine the one-dimensional energy dissipation rate. These measurements and the dissipation rate calculated from them helped to

validate the measurements conducted with the WindScanner. The dissipation rate is determined with Equation 6 for an isotropic homogeneous flow, described by Tennekes and Lumley (1972), Hinze (1975) or Monin and Yaglom (1971). Here $\nu$ is the kinematic viscosity of the flow, $\sigma_u$ is the variance of the flow and $\lambda$ is the Taylor-micro length scale.

$$\varepsilon = 15\nu \left\langle \left( \frac{\partial u'}{\partial x} \right)^2 \right\rangle = 15\nu \frac{\sigma_u^2}{\lambda^2} \tag{6}$$



The Taylor-micro length scale is determined with Equation 7 using the energy spectrum over the wave number $k$ ($E(k) = E(f)u/2\pi$), where $f$ is the frequency and $u$ is the mean velocity. The term $\left\langle \left(\partial u'/\partial x\right)^2 \right\rangle$ is determined by integrating the energy spectrum (Equation 8) between the minimum wave numbers ($k_{\min}$) and the large wave numbers ($k_{\max}$), above where the turbulence only contains artifacts (eg., due to the measurement system) but no flow events according to Neunaber (2019).

$$\lambda = \left( \frac{\sigma^2}{\left\langle \left(\partial u'/\partial x\right)^2 \right\rangle} \right)^{1/2} \tag{7}$$

$$\left\langle \left( \frac{\partial u'}{\partial x} \right)^2 \right\rangle = \int_{k_{\min}}^{k_{\max}} k^2 E(k)dk \tag{8}$$

## 3 Results and Discussion of Curled Wake Measurements

The results for the flow field within the empty test section at each inflow condition are presented in Section 3.1. Then the effects of the inflow conditions and operational settings (yaw angle) on the wake characteristics are analyzed. The temporally averaged streamwise velocity results, the turbulent kinetic energy dissipation rate, the wake shape and the wake deflection are presented each in a subsequent section.

### 3.1 Undisturbed Flow Propagation Through the Test Section

In order to conduct a meaningful analysis of wake characteristics a well-defined flow field is required, avoiding that the turbulence and wind shear characteristics break down within the measurement domain. This was verified by measuring the flow in the empty test section for all three inflow conditions. The measurements were analyzed by spatially averaging the mean streamwise velocity component at each height ($z$) for all values of horizontal position ($y$) between -1,5 $D \leq$ y $\leq$ 1,5 $D$. The mean flow between 1 $D$ and 16 $D$ for the case without grid is presented in Figure 2a. The mean flow between 0 $D$ and 10 $D$ for the cases with a uniform passive grid and the sheared (boundary layer) inflow is shown in Figure 2b and Figure 2c, respectively.

The uniformity of the flow for the inflow condition without grid (Figure 2a) confirms the stability of the flow throughout the measurement domain. Furthermore, the boundary layer developing on the wind tunnel floor appears unlikely to influence turbine wake development, as it is grows from ≈0.2 m at 1 $D$ to ≈0.4 m at 16 $D$, which remains well below the rotor area (highlighted as a shaded green area).

The mean flow for the case with a uniform passive grid also indicates a stable condition (Figure 2b). A similar growth of the boundary layer, from ≈0.2 m at 1 $D$ to ≈0.48 at 10 $D$, is observed (Figure 2c). The boundary layer appears to grow slightly faster than for the case without grid due to the increase in the turbulence intensity which increases mixing between the boundary layer and the steady flow.

The mean flow for the case of the passive grid with variable open area does not show a well-behaved boundary layer close to the wall (Figure 2c), due to a slight speed-up of the mean wind speed between 0.05 m $\leq z \leq$ 0.2 m. A possible cause of the





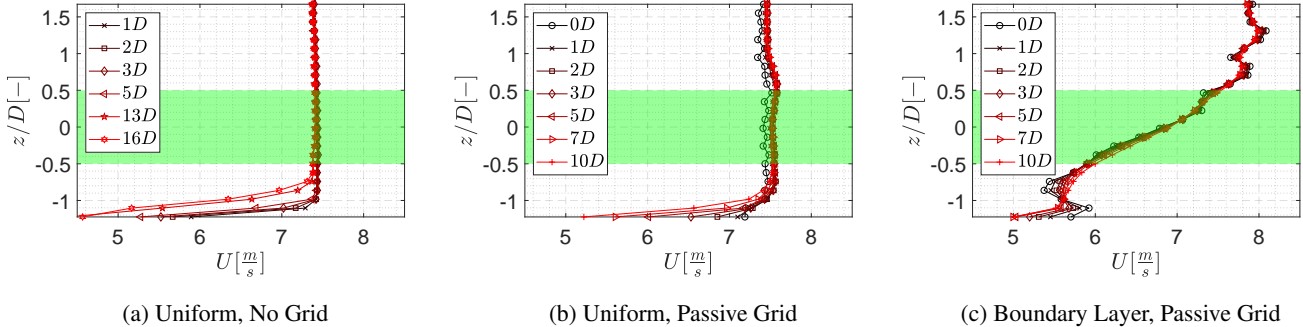

| (a) Uniform, No Grid | (b) Uniform, Passive Grid | (c) Boundary Layer, Passive Grid |

**Figure 2.** Mean velocity $U[\frac{m}{s}]$ spatially averaged at each height position $z$ for all values of $y$ between -1,5 $D \leq y \leq 1,5\,D$, extracted from the measurements within vertical planes at multiple downstream positions. **Green:** Rotor area

speed-up is the mounting of the active grid frame between the wind tunnel nozzle and the first test section element. As it was not possible to completely seal the connection between the active grid and the wind tunnel segment, air from the outside was sucked into the wind tunnel segment near the bottom leading to the visible speed-up region. However, the flow-field within the rotor area showed a near constant shear exponent (power law fitted over the vertical extent of the rotor area), with a variation

5    between $\alpha$ =0.28 to $\alpha$ =0.27 within the rotor area. The slight variation is considered to have a negligible effect on the wake characteristics. This indicates that flow is stable and provides a reasonable approximation of an atmospheric boundary layer flow and is suitable for the investigation of turbine wake characteristics.

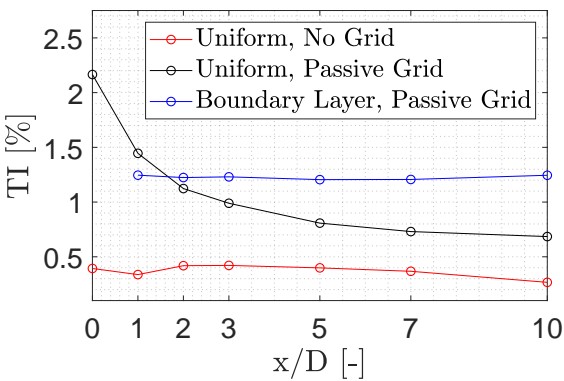

**Figure 3.** Turbulence intensity for each inflow condition within the wind tunnel segments at each downstream distance at hub height conducted with the staring mode measurements.

In addition to the mean flow, turbulence characteristics is crucial to determine the stability of the flow. The analysis was conducted using the staring mode measurements at hub height at the centre of the wind tunnel, and the results are shown

10   in Figure 3. As expected the turbulence intensity remains approximately constant around 0.37% for the case without grid (empty nozzle). For the inflow condition with a uniform passive grid, the turbulence is higher initially and the rapidly decays



moving downstream and stabilizes after $5\,D$. Groth and Johansson (1988) indicated that the turbulence length scale decays behind a passive grid and thus leads to the reduction of the turbulence intensity. The effect of the turbulence decay on the wake characteristics is considered to be minor as the turbulence intensity reduces from 1.7% to 0.7%. For the inflow condition with the variable open area grid (boundary layer), the turbulence intensity remains approximately constant around 1.5%. This

indicates that turbulence is maintained at a near constant level due to the mean shear, which causes production of turbulent kinetic energy. This reduces the generated turbulence length scale and thus leads to a faster decay of the turbulence intensity. (Note that the measurement at $0\,D$ is not included for this inflow condition due to a faulty data set.)

## 3.2 Wake Characteristics at Different Operational and Inflow Conditions

Through the use of multiple scanning planes with the WindScanner the evolution of the wake can be analysed in detail for each

inflow condition. As representative locations, the measurements at $2\,D$ and $5\,D$ will be compared for the nine combinations of inflow and turbine operating conditions. By $2\,D$, tip and hub vortices have broken down and the wake is transitioning out of its near-wake character. By $5\,D$ the wake has achieved far-wake character (as defined for wind turbines), while still exhibiting a pronounced wake signature.

Figure 4 shows the wake characteristics at $2\,D$ downstream from the turbine for each inflow condition and for the yaw angles

of $\psi = -30°, 0°, 30°$. This figure highlights the symmetric behaviour, at $\psi = 0°$, and the asymmetric behaviour of the wake, at $\psi = \pm 30°$. Furthermore, the development of the curled wake due to the CVP described by Bastankhah and Porté-Agel (2016) is visible. The curled wake is visible for the case with a boundary layer inflow, while the wake has an elliptical shape for the cases with uniform inflow.

Next to the development of the curled wake, the wake of the tower is also visible in Figure 4. Here it can be detected that

the tower wake experiences a lateral displacement in the opposite direction of the wake deflection, with no lateral movement for the case of $\psi = 0°$. This is a result of conservation of mass, balancing the lateral velocity component at hub height created by the CVP with a lateral momentum in the opposite direction at a large yaw angle. This may be an effect of conducting the experiment in a closed test section and should be investigated further.

Furthermore, Figure 4a and Figure 4c show an asymmetric distribution of the wind speed within the wake. For a yaw angle

of $\psi = -30°$ (Figure 4a) the upper part of the wake area has a lower wind speed in comparison to the lower part of the wake. Due to the large yaw angle and the clockwise rotation of the turbine, the blades of the turbine will either rotate into or with the wind, thus changing the relative wind speed and the angle of attack. In the case with $\psi = -30°$ (Figure 4a) the blades turn into the wind in the upper part of the rotor area, increasing the relative wind speed.



(a) Uniform, No Grid, $\psi = -30°$

(b) Uniform, No Grid, $\psi = 0°$

(c) Uniform, No Grid, $\psi = 30°$

(d) Uniform, Passive Grid, $\psi = -30°$

(e) Uniform, Passive Grid, $\psi = 0°$

(f) Uniform, Passive Grid, $\psi = 30°$

(g) Boundary Layer, Passive Grid, $\psi = -30°$

(h) Boundary Layer, Passive Grid, $\psi = 0°$

(i) Boundary Layer, Passive Grid, $\psi = 30°$

**Figure 4.** Temporally averaged streamwise velocity component at a downstream distance of $2\,D$ for different inflow conditions. The wake is viewed looking upstream towards the turbine model.





**Figure 5.** Temporally averaged streamwise velocity component at a downstream distance of 5 $D$ for different inflow conditions. The wake is viewed looking upstream towards the turbine model.



The opposite occurs with $\psi = 30°$ (Figure 4c). In addition, the difference between the wind speed in the upper part and the lower part reduces for the case with a uniform passive grid shown in Figures 4d to 4f. This is related to the higher turbulence intensity and the higher mixing rate. Figures 4g to 4i indicate that the lowest wind speed region is approximately at hub height.

At 2 D with $\psi = -30°$ (Figure 4g) the region with the lowest wind speed propagates to the lower part of the wake area. The opposite occurs at $\psi = 30°$ (Figure 4i). In addition, it is also visible that the shear layer of the wake with the surrounding flow is less severe in comparison to the other inflow cases, due to the higher turbulence intensity.

Figure 5 shows the wake characteristics at 5 D downstream from the turbine for each inflow condition and for the yaw angles of $\psi = -30°, 0°, 30°$, 3 D further downstream than Figure 4. At this distance the wake has evolved to a curled wake shape for each inflow condition for $\psi = \pm 30°$, indicating that the lateral displacement induced by the CVP increased further downstream. This is consistent with the findings of Bastankhah and Porté-Agel (2016). In addition, with the increased lateral displacement of the turbine wake at hub height, the wake of the tower also experienced a larger lateral displacement in the opposite direction for the cases with $\psi = \pm 30°$, as expected. Furthermore, the wake experiences mixing between the boundary of the wake and the ambient flow (Sanderse (2009)). Here, it can be seen that the mixing layer between the wake and the free-stream velocity is smaller at the inflow condition with no grid in comparison to the inflow condition with a passive grid and a boundary layer. This is due to the higher turbulence intensity in the latter cases. As expected the wake mixing layer grows for all cases as the wake evolves downstream.

The region with the highest wake deficit is transferred to a certain direction depending on the yaw angle, highlighted in Figure 6. This is due to the combination of the wake rotation, the CVP and the increase of the rotation strength further downstream described in Martínez-Tossas et al. (2019). The comparison between Figure 4 and Figure 5 indicate that the wake deficit transferred to the lower part of the wake area for $\psi = -30°$ and to the upper part for $\psi = 30°$ for each inflow case. The transfer of the wake deficit is visualized in Figure 6 for each inflow case at each downstream distance. The centre is determined with the normalized local velocity ($\frac{u_{i,\infty} - u_i}{u_{i,\infty}}$) by calculating the centroid over the region below the threshold $\frac{u_{i,\infty} - u_i}{u_{i,\infty}} > max(\frac{u_{i,\infty} - u_i}{u_{i,\infty}}) - \frac{u_{thresh}}{u_{i,\infty}}$, $u_{thresh} = 0.25 \frac{m}{s}$. It can be seen that the deficit is transferred with the counter-clockwise rotation of the wake when looking downstream. This has also been described in Bastankhah and Porté-Agel (2016) showing the displacement of the wake centre predicted by the potential flow theory. It should be noted that there is some scatter in Figure 6a for the position of the wake deficit at 13 D and at 16 D, since the wake velocity deficit has further decayed which makes the wake centre detection detection methods less accurate for the given measurement resolution.





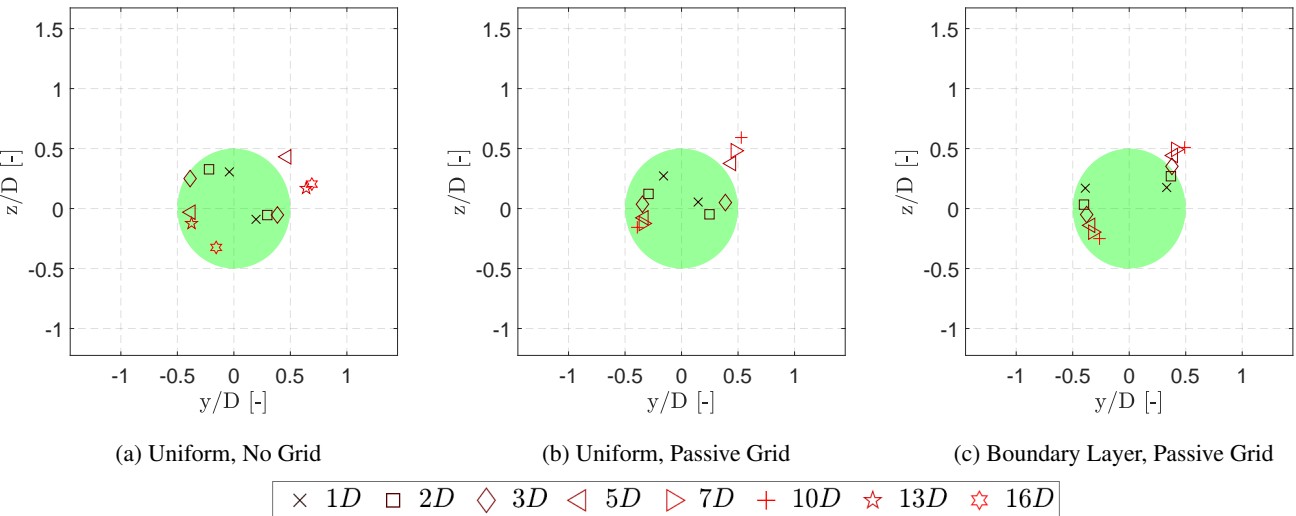

(a) Uniform, No Grid       (b) Uniform, Passive Grid       (c) Boundary Layer, Passive Grid

$$\times\ 1D \quad \square\ 2D \quad \lozenge\ 3D \quad \triangleleft\ 5D \quad \triangleright\ 7D \quad +\ 10D \quad \star\ 13D \quad \star\ 16D$$

**Figure 6.** Location of the largest mean velocity deficit normalized over the local velocity $(\frac{u_i - u_{i,\infty}}{u_{i,\infty}})$ for each inflow condition at each downstream position for $\psi = \pm 30°$. At $\psi = -30°$ the positions are located at $y < 0$. At $\psi = 30°$ the positions are located at $y > 0$.

### 3.3 Evolution of the Energy Dissipation Rate

The turbulent kinetic energy dissipation rate ($\varepsilon$) gives an indication how the flow behaves within the wake: a high dissipation rate indicates a faster mixing of the wake whereas a small dissipation rate suggests that the wake will persist further downstream. Figure 7 indicates the estimated $\varepsilon$ (Equation 5) of the wake at a downstream distance of $2\,D$. The first noticeable

5   difference is that the energy dissipation rate is slightly lower in the case with no grid (Figure 7a to 7c) in comparison to the situations with a passive grid or a boundary layer, as expected. This is related to the higher turbulence in the inflow in the latter two conditions.

Moreover, an enhanced dissipation rate was detected within the wake in comparison to the ambient flow, which corresponds to Lundquist and Bariteau (2015). Furthermore, for $\psi = 0°$ a ring-shaped area of enhanced dissipation slightly larger than

10   the rotor area exists for each inflow condition, which overlaps with the wake mixing layer and is consistent with the findings of Eriksen and Krogstad (2017), Bartl and Sætran (2017) and Schottler et al. (2018). In addition, this is also related to the wake generated by the tower. Moreover, the energy dissipation rate captured within the tower wake reduces with a sheared inflow. The high dissipation rate at the wake centre can be related to the root vortices within the near wake. At $\psi = \pm 30°$ the circular shape of the energy dissipation is deflected to an elliptical shape with a uniform inflow or a curled shape with a sheared inflow.

15   A region with a low dissipation rate is visible within the wake area in the case with no grid, a passive grid and a boundary layer, corresponding to the region with the highest wake deficit which also has comparatively lower velocity gradients. An enhanced energy dissipation rate is also visible in the upper part of the wake with a sheared inflow (Figure 7g to 7i), which is a result of the higher velocity components resulting in higher mixing rate between the ambient flow and the wake.





**Figure 7.** Energy dissipation rate within the wake at a downstream distance of $2\,D$ for different inflow conditions. The wake is viewed looking upstream towards the turbine model.







**Figure 8.** Energy dissipation rate within the wake at a downstream distance of $5\,D$ for different inflow conditions. The wake is viewed looking upstream towards the turbine model.





At $5\,D$ (Figure 8) the energy dissipation rate decreased due to the expansion of the mixing area between the wake and the ambient flow. Furthermore, the width of the ring visible at $\psi = \pm 30°$ increased at $5\,D$, which is also related to the enhanced mixing at a larger downstream distance. This also caused that the increment of the energy dissipation rate at hub height is not visible at a downstream distance of $5\,D$. A similar behaviour is noticeable at the upper part of the wake with sheared inflow

condition indicating an enhanced energy dissipation rate in comparison to the lower part of the wake area. Due to the large yaw angle and the downstream distance, the circular shape of the energy dissipation rate has deformed to a curled shape for each inflow condition. In addition, similar to the flow at $2\,D$ a low dissipation rate is also visible at the region with the highest wake deficit.

The energy dissipation is larger in the uniform inflow cases in comparison to the sheared cases at $5\,D$. This could be related

to the enhanced mixing with a sheared inflow between the regions with a higher wind speed in comparison to the regions with a lower wind speed, resulting in a faster decay of the flow. This is visible in Figure 7f - 7d showing a larger region with a high energy dissipation rate with a sheared inflow at $2\,D$ in comparison to the uniform inflow cases. This indicates that the wake will break down faster in a boundary layer, whereas the wake deficit persists for a longer period of time with a uniform inflow condition.

## 3.4  Development of the Curled Wake Shape

Figure 9 visualizes the growth of the curled wake behind the turbine. The contour-lines indicate the boundary of the wake determined by implementing a threshold to the normalized local velocity with the free stream velocity. The threshold is set at $\frac{u_{i,\infty} - u_i}{u_{i,\infty}} = 0.9$. The formation of a curled wake shape is due to the CVP described by Bastankhah and Porté-Agel (2016), leading to a lateral velocity component at hub height and two vortex pairs, rotating in the opposite direction, at the top and

bottom of the rotor area depending on the yaw angle. This was also observed by Vollmer et al. (2016) at different atmospheric conditions and was implemented in the FLORIS model by Martínez-Tossas et al. (2019).

It can be seen in Figure 9a and in Figure 9b, that the wake area has an elliptical shape in the near-wake region and then evolves to a curled shape between $3\,D$-$5\,D$ for $\psi = \pm 30°$ for the inflow condition with a uniform passive grid. The inflow condition without a grid showed a similar trend (not shown). For the case with a boundary layer inflow (Figure 9c and Figure 9d), the

wake area evolved sooner to the curled shape between $2\,D$-$3\,D$. This can also be seen in Figure 10 which plots the $y$-position of the minimum wake velocity (largest velocity deficit) for each $z$-position. In order to determine the largest velocity deficit the measurement data shown in Figure 4 and in Figure 5 are smoothed to remove local fluctuations. This resulted in arc-shaped curves which provide a direct comparison of the wake shape for different inflow conditions, similar to Bartl et al. (2018). It can be seen that for the case with a uniform inflow the curve is almost a straight line at $2\,D$ for $\psi = \pm 30°$ and slowly shapes

to an arc-shaped curve at $3\,D$. In the case with a boundary layer inflow the curve is already arc-shaped at $1\,D$ for $\psi = \pm 30°$. This indicates that the combination of the shear layer, the wake rotation and the CVP increases the lateral velocity at hub height induced by the CVP, thus increasing the vortex strength of the CVP in comparison to a uniform case. In addition, at $\psi = -30°$ the wake experiences a larger deflection of the minimum velocity with a boundary layer inflow in comparison to the case with a uniform inflow, which corresponds with the presence of a stronger CVP. Moreover, the maximum deflection





of $\pm 0.72\,D$ is symmetric for the inflow with a uniform passive grid for $\psi = \pm 30°$. For a boundary layer inflow the maximum deflection is slightly asymmetric with $0.72\,D$ for $\psi = 30°$ and $-0.81\,D$ for $\psi = -30°$. This is related to the combination of the counter-clockwise rotation of the wake and the sheared inflow, leading to an increase or a decrease of the relative velocity and hence the vortex strength depending on the yaw angle as described in Section 3.2.

5  ### 3.5    Wake Deflection at Different Inflow Conditions

The influence of the different inflow conditions and operational conditions on the deflection of the wake is further analysed here using the methods described in Section 2.3. Figure 11 illustrates the wake centre for each inflow condition computed with the Gaussian function and the minimal power (Section 2.3). The shaded area indicates the spread ($\pm\sigma$) of the wake centre derived from the individual scans with both methods. It is clear that the Gaussian-based method results in large wake deflections in all

10  yawed cases, since it is influenced by the location of the largest velocity deficit, whereas as the method determining the minimal

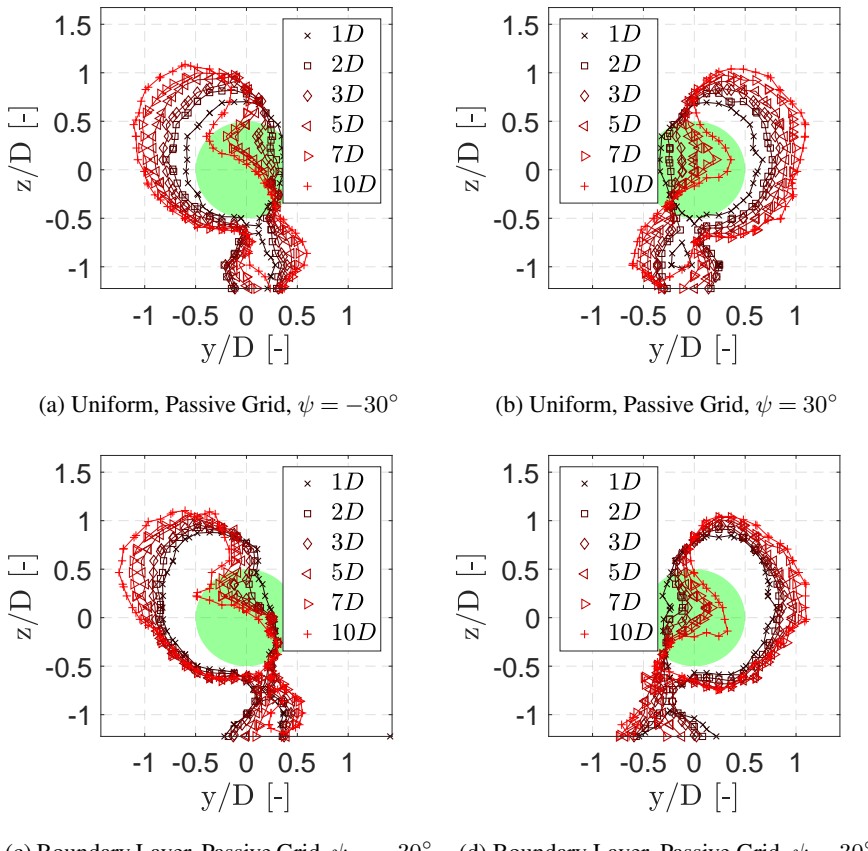

**Figure 9.** Growth of the curled wake behind the wind turbine model at multiple downstream distances. The contour-lines indicate the boundary with $\frac{u_{i,\infty} - u_i}{u_{i,\infty}} = 0.9$. The view is looking upstream towards the turbine model.



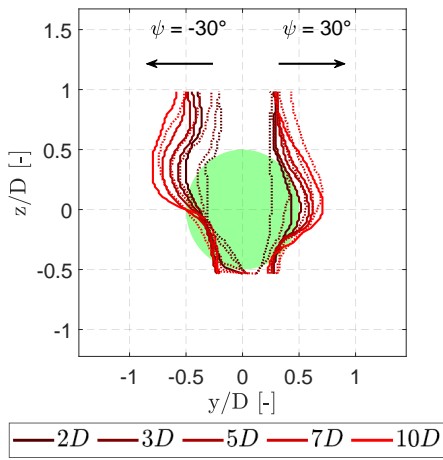

**Figure 10.** Position of the minimum velocity determined at each $z$ position for each inflow condition. **Dotted:** Uniform, Passive Grid **Solid Lines:** Boundary Layer, Passive Grid

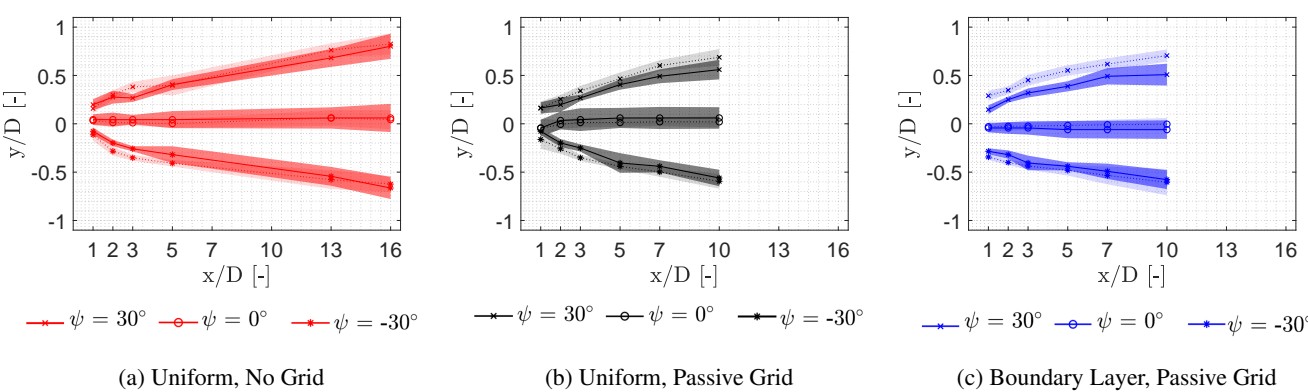

(a) Uniform, No Grid          (b) Uniform, Passive Grid          (c) Boundary Layer, Passive Grid

**Figure 11.** Wake deflection at different inflow condition determined with the methods described by Schottler et al. (2018). The wake centre is derived from the vertical scans at each downstream distance. The shaded area indicate the spread of the wake centre determined for each individual vertical scan. **Dotted lines:** Gaussian method **Solid lines:** Minimal potential power method

potential power accounts for the overall wake area. The difference is also related to the development of the curled shape, shown in Figure 10 illustrating an arc-shaped curve of the highest velocity deficit. The difference between the two methods for finding the wake centre can clearly be seen for the case with a boundary layer inflow and $\psi = 30°$, where the wake is curled the most.

The asymmetry in the wake deflection, observed by Fleming et al. (2014), Fleming et al. (2018) and Bartl et al. (2018), is
5    visible in Figure 11 for the case with a boundary layer inflow. Using the method by locating the minimal potential power the wake centre experiences a larger wake deflection for $\psi = 30°$ in comparison to $\psi = -30°$. This is due to the influence of the shear on the strength of the vorticity on the CVP and corresponds with the findings shown in Figure 10.





## 4 Validation of Wake Characteristics of Undeflected Wake Measurements

The velocity data acquired with the WindScanner was validated by comparing it to the hot-wire data set from Neunaber (2019), shown in Section 4.1. In that study the wake characteristics were determined through multiple hot-wires with the same layout, wind turbine model, inflow condition and operational condition similar to the case with no grid and $\psi = 0°$ in the present
campaign. The comparison is followed by an uncertainty analysis of the measurement data in Section 4.2.

### 4.1 Data Comparison

Neunaber (2019) used the MoWiTO 0.6 to determine the wake characteristics at multiple downstream and lateral positions through the use of hot-wires. Due to the complex nature of the near wake region, temporally averaged properties of the flow obtained from the staring mode measurements (f = 451.7 Hz) from the WindScanner with $\psi = 0°$ are compared with the hot-
wire measurements (f = 15 kHz) at $5\,D$ and at $10\,D$. A smaller downstream distance is not used for the comparison, since a small misalignment of the measurement position could lead to a large velocity difference. Furthermore, the comparison of the temporally averaged properties of the flow is assessed by performing a visual comparison of the horizontal wake development and the vertical scans at $1\,D$, $2\,D$, $3\,D$ and $5\,D$.

| | Neunaber (2019) | | WindScanner Campaign | |
|---|---|---|---|---|
| $\frac{x}{D}$ | $u_{\mathrm{mean}}\left[\frac{m}{s}\right]$ | $\sigma_{\mathrm{mean}}\left[\frac{m}{s}\right]$ | $u_{\mathrm{mean}}\left[\frac{m}{s}\right]$ | $\sigma_{\mathrm{mean}}\left[\frac{m}{s}\right]$ |
| 5 | 2.05 | 0.35 | 1.70 | 0.49 |
| 10 | 4.54 | 0.62 | 4.81 | 0.49 |

**Table 2.** Comparison of $u_{\mathrm{mean}}$ and $\sigma_{\mathrm{mean}}$ at $5\,D$ and at $10\,D$ on wake centre line obtained by the hot-wire measurements (Neunaber (2019)) and in the WindScanner campaign with the staring mode measurements. Both measurements were conducted with no grid at $U = 7.5\frac{m}{s}$ at hub height.

The temporally averaged properties of the flow behind the turbine are shown in Table 2. Differences exist both for the mean
wind speed and the standard deviation between the two measurement campaigns. These can be attributed to the location of the staring mode measurements in relation to the position of the turbine, as a small deviation can lead to a large difference, and the much larger sampling and averaging volume of the WindScanner measurements, which can casue spatial averaging of the turbulence. For example in the hotwire measurement campaign by Neunaber (2019), a shift of $0.21\,D$ (the lateral spacing) would lead to a mean wind speed of $u_{\mathrm{mean}} = 2.53\frac{m}{s}$ and a standard deviation of $\sigma = 0.66\frac{m}{s}$. Furthermore, the difference in
the sampling frequency and the spatial averaging by the probe volume also effects the temporally averaged properties of the flow.

To provide a more insightful comparison of the data, the hot-wire data is compared with the horizontal scan measured by the WindScanner, shown in Figure 12. Here the lines with a marker indicate the contours for wind speeds of $U = [3, 4, 5, 6, 7]\frac{m}{s}$. The wake measured with the horizontal scan by the WindScanner shows a similar development as the wake measured by Ne-





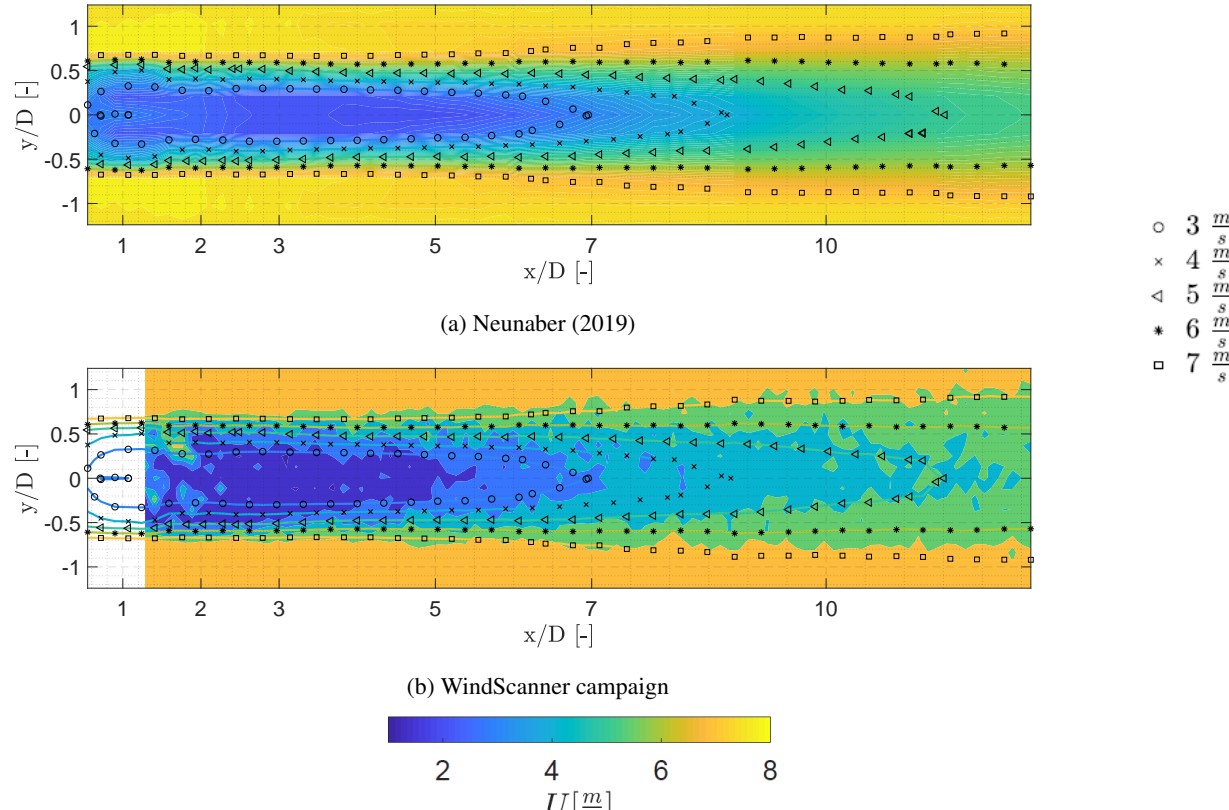

(a) Neunaber (2019)

(b) WindScanner campaign

**Figure 12.** Comparison of the profiles of the averaged streamwise velocity component at $1\,D$, $2\,D$, $3\,D$ and $5\,D$ between the WindScanner measurements and the hot-wire measurements at Neunaber (2019). Both measurements were conducted with no grid at $U = 7.5\frac{m}{s}$. Lines with a marker indicate the contours at $U = [3, 4, 5, 6, 7]\frac{m}{s}$.

unaber (2019) with regards to wake growth and position of the transition of the near wake to the far wake.

Furthermore, the temporally averaged streamwise velocity component at hub height from the hot-wire measurements were compared with the vertical Lidar scans at $1\,D$, $2\,D$, $3\,D$ and $5\,D$. At $1\,D$ in Figure 12 it can be seen that the wake shape obtained from the WindScanner resembles the wake deficit obtained by the hot-wire measurements. However, there is a large difference in the streamwise velocity component around $y/D = 0$. This is due to the filtering process of the WindScanner, since the spectrum of the WindScanner is heavily disturbed due to the nacelle of the wind turbine resulting in a low amount of measurement samples at that grid point. At $2\,D$ to $5\,D$ the wake profile shows a similar wake deficit and wake width. However, small differences are noticeable in the velocity. This can also be related to the filtering process and the effect of the sampling onto the grid. Furthermore, the probe volume length is $20\,\text{cm}$ at $1\,D$ and $13\,\text{cm}$ at $5\,D$ which also influences the measurement data.





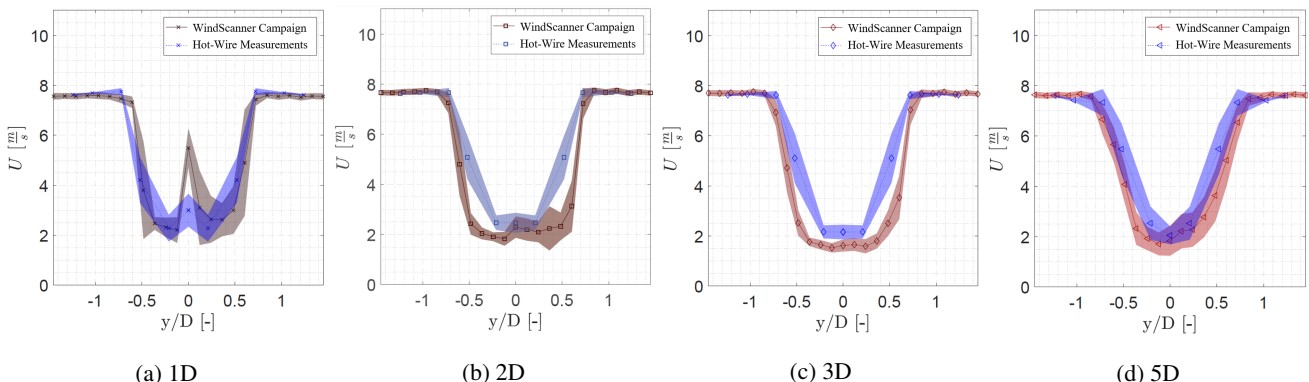

|   |   |   |   |
|---|---|---|---|
| (a) 1D | (b) 2D | (c) 3D | (d) 5D |

**Figure 13.** Comparison of the profiles of the averaged streamwise velocity component at $1\,D$, $2\,D$, $3\,D$ and $5\,D$ between the WindScanner measurements and the hot-wire measurements at Neunaber (2019). Both measurements were conducted with no grid at $U = 7.5\frac{m}{s}$. The shaded area shows the $\pm\sigma$ of the measurement data.

Next, the energy dissipation rate acquired from the staring mode measurements with the WindScanner was compared to the energy dissipation rates measured with the hot-wires at $10\,D$ behind the turbine. This resulted in energy dissipation rates of $\varepsilon = 1.56\frac{m^2}{s^3}$ and $\varepsilon = 1.31\frac{m^2}{s^3}$, respectively. Figure14 indicates that the development of the energy dissipation rate within the wake is similar in comparison to the data obtained by Neunaber (2019), showing a low energy dissipation rate within the near

wake region and outside of the wake. The dissipation rate reaches its highest value near the edges of the wake in regions of high shear. Further downstream the energy dissipation rate slowly reduces. This agrees with Figure 7b and Figure 8b, showing a ring with a high energy dissipation rate slightly larger than the rotor area which increases at $5\,D$ in width. Furthermore, the energy dissipation at the rotor centre increases up to 5-7 $D$, after which it reduces. This also agrees with Figure 7b and Figure 8b showing an increase of the energy dissipation rate at the rotor centre.

The comparison of the energy dissipation rates between the two measurement techniques is further expanded in Figure 15, showing the energy dissipation rate at hub height. A similar trend is visible, showing a high energy dissipation rate at $y/D = 0$ at $1\,D$ and at the shear layer between the wake and the freestream flow at $2\,D$ to $5\,D$. It can be seen that the energy dissipation rate obtained from the WindScanner is lower at $1\,D$ in comparison to the hot-wire measurements, which could be due to the backscatter of the blades effecting the measurements. However, at $2\,D$ to $5\,D$ it is noticeable that the magnitude of the energy

dissipation rate obtained with the WindScanner differs in comparison to the hot-wire measurements. This can be related due to the probe volume crossing the entire wake and the difference in sampling frequency. Furthermore, the difference of the mean velocity gradient is not accounted for with the method used to determine the energy dissipation. Another influence is the sampling frequency and the method used to calculate the energy dissipation rate from the hot-wire measurements, where the energy spectrum has been cut-off at a certain wave number to exclude artefacts in the turbulence.

In addition, the measurements obtained with the hot-wires during the WindScanner campaign indicate a similar magnitude of the streamwise velocity and energy dissipation rate with the WindScanner measurements shown in Figure 16. The temporal averaged velocity, shown in Figure 5i, indicates a similar curvature of the wake (**Red Crosses**) in comparison to the





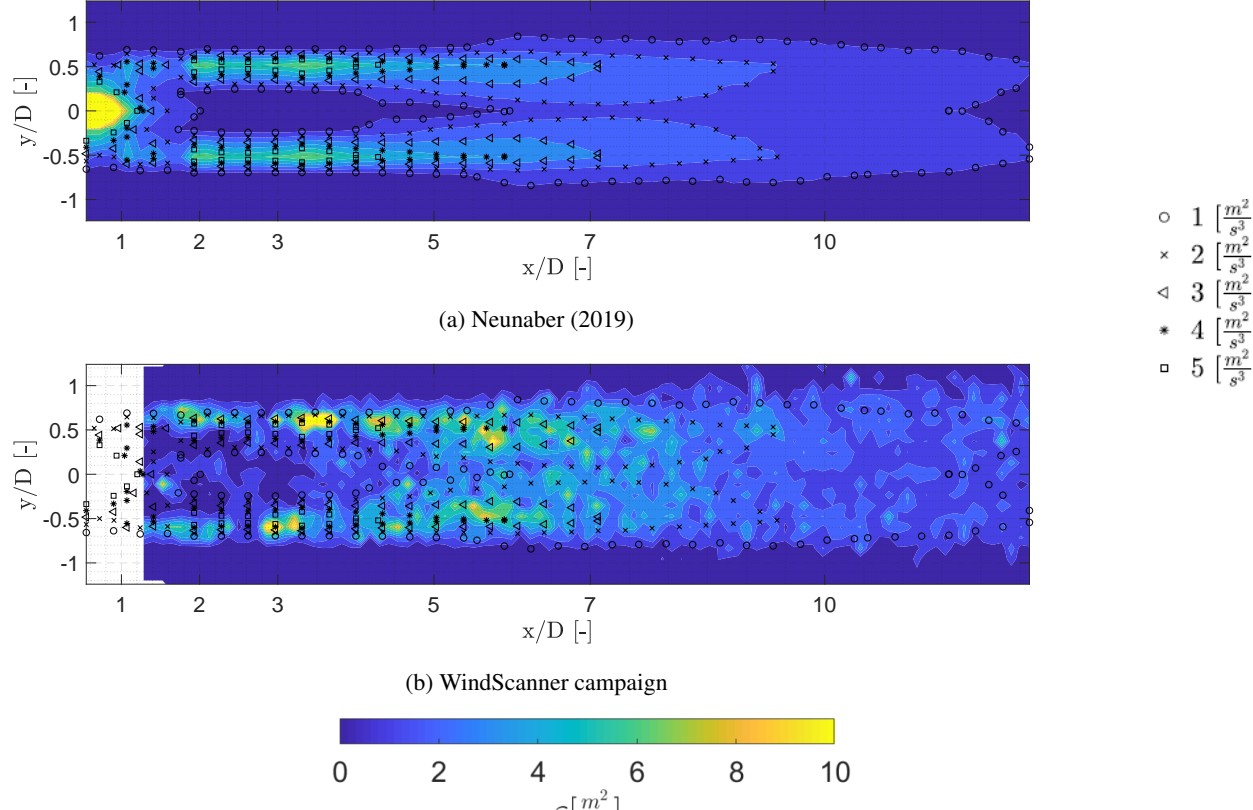

**Figure 14.** Comparison of the profiles of the averaged energy dissipation rate at $1\,D$, $2\,D$, $3\,D$ and $5\,D$ between the WindScanner measurements and the hot-wire measurements at Neunaber (2019). Both measurements were conducted with no grid at $U = 7.5\,\frac{m}{s}$. Lines with a marker indicate the contours at $U = [0.6, 1.2, 1.8]\,\frac{m}{s}$

WindScanner data (**Blue Crosses**). Similar to the WindScanner data (Figure 8i), a higher dissipation rate is also observed with the hot-wire measurements (Figure 16b) at the upper region of the wake ($\varepsilon \approx 3.8\,\frac{m^2}{s^3}$) in comparison to the lower region ($\varepsilon \approx 1.5\,\frac{m^2}{s^3}$). Moreover, within the ambient air, an energy dissipation rate of $\varepsilon \approx 0.28\,\frac{m^2}{s^3}$ is observed which corresponds to the measurements with the WindScanner. Furthermore, the energy dissipation rate has a similar magnitude as the one obtained with

5  the WindScanner in the upper region of the wake, while in the lower region of the wake a difference is noticeable. This could be due to the probe volume averaging of the WindScanner, as the volume (13 cm at $5\,D$) crosses the entire wake leading to unwanted artefacts in the spectrum. This indicates that the WindScanner is able to capture the main trends of the development of the energy dissipation rate but that it is effected due to the probe volume averaging.



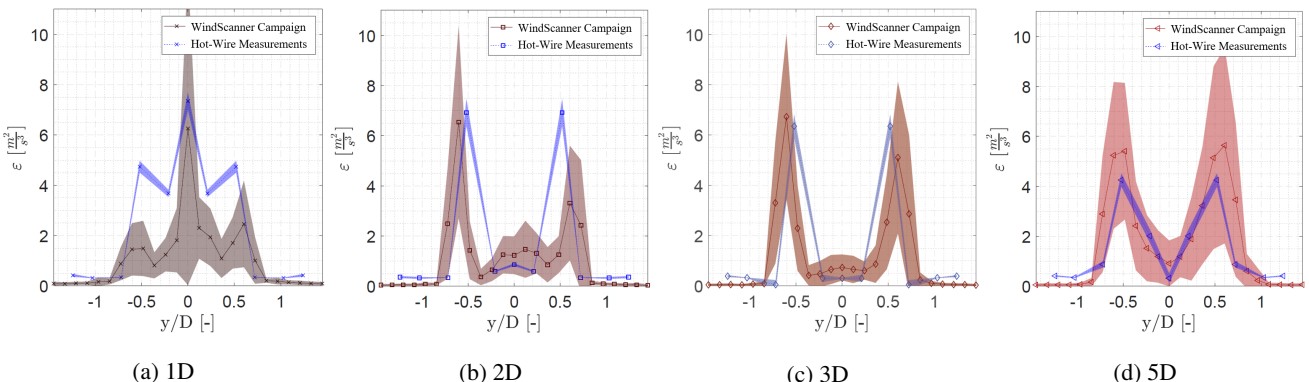

| (a) 1D | (b) 2D | (c) 3D | (d) 5D |

**Figure 15.** Comparison of the profiles of the averaged energy dissipation rate at $1\,D$, $2\,D$, $3\,D$ and $5\,D$ between the WindScanner measurements and the hot-wire measurements at Neunaber (2019). Both measurements were conducted with no grid at $U = 7.5\,\frac{m}{s}$. The shaded area shows the $\pm\sigma$ of the measurement data.

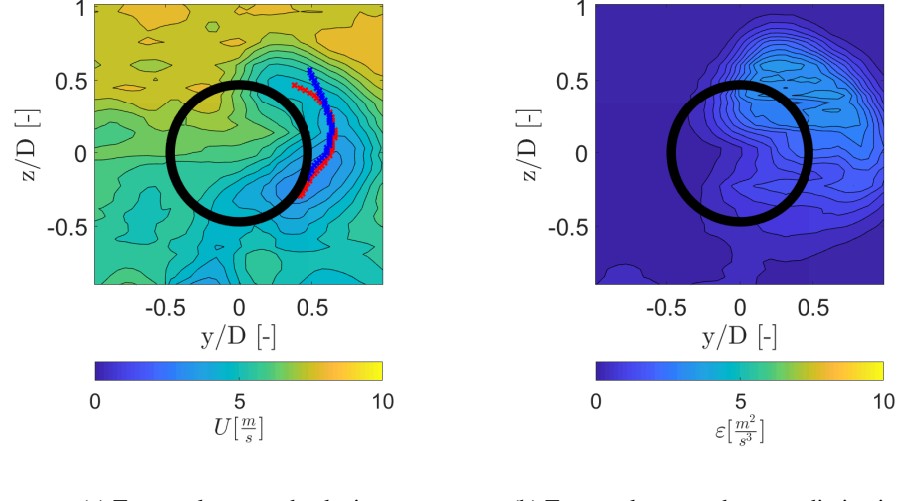

| (a) Temporal averaged velocity | (b) Temporal averaged energy dissipation rate |

**Figure 16.** Temporal averaged velocity (a) and energy dissipation rate (b) obtained from the hot-wire measurements conducted in the WindScanner campaign. The position of the minimum velocity determined at each $z$ position, which indicates the curvature of the wake centre, obtained from the WindScanner (**Blue Crosses**) and the hot-wire (**Red Crosses**) are visualized

## 4.2 Uncertainty Analysis

The uncertainty analysis for the estimation of the $u$-component is conducted following the standard uncertainty method performed by Stawiarski et al. (2013) and van Dooren et al. (2016). The investigation was conducted by considering the uncertainty of the Lidar measurement and the reconstruction for the streamwise velocity component at a downstream distance of $2\,D$ and $5\,D$, computed using Equation 2, to determine the uncertainties of the horizontal velocity component $e_u$. In Equa-



tion 9 $e_\text{v}$ and $e_\text{w}$ are the uncertainty of the $u$- and $w$-component, $e_\delta$ and $e_\theta$ are the uncertainty of the azimuth and elevation angle and $e_\text{vlos}$ is the uncertainty of the measured line-of-sight velocity, which is assumed to be 1% according to Pedersen et al. (2012). In addition, the maximum uncertainty of the $w-$ and $v-$component is set to be $1\frac{m}{s}$ at $1\,D$, $5\,D$ and $10\,D$. This is due to the strong vortices within the near wake. Furthermore, the pointing error is assumed to be 0.05 mrad for the elevation and

the azimuth angle (van Dooren et al. (2017)).

$$e_\text{u}^2 = (\frac{\partial u}{\partial V_\text{LOS}} e_{V_\text{LOS}})^2 + (\frac{\partial u}{\partial v} e_\text{v})^2 + (\frac{\partial u}{\partial w} e_\text{w})^2 + (\frac{\partial u}{\partial \delta} e_\delta)^2 + (\frac{\partial u}{\partial \theta} e_\theta)^2 \qquad (9)$$

Figure 17 presents the error obtained with Equation 9 for the cases $2\,D$ and $5\,D$ for the inflow condition without a grid normalized with the horizontal velocity component at a certain position ($\frac{e_\text{u}}{u}$ [%]). At a downstream distance of $2\,D$, the error is between 1 % in the ambient air and 3.1 % within the wake. This is expected since the line-of-sight velocity is lower within the

wake. In addition, the error is larger at the upper part of the measurement domain ($\frac{z}{D} > 1$) in comparison to the lower region ($\frac{z}{D} < -1$), due to the larger elevation angle of the laser beam. This leads to an error of 1.5 % in the upper region and 1 % in the lower region at $2\,D$ and $5\,D$. Within the wake the error ranges between 3.1 % and 2.6 % at $2\,D$ and $5\,D$ respectively.

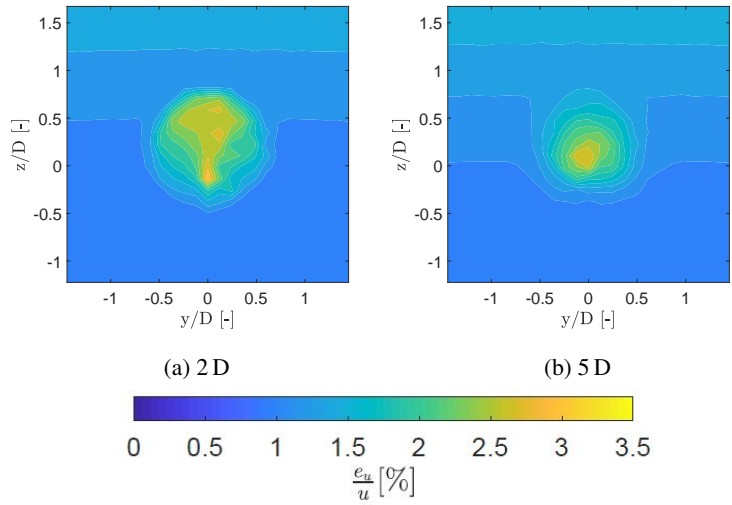

(a) 2 D          (b) 5 D

**Figure 17.** Normalized error $\frac{e_\text{u}}{u}$ [%] at 2 D and 5 D for the inflow without a grid.

Next, the statistical uncertainty needs to be considered using the margin of error $e_\text{MOE} = z_\gamma \sigma / \sqrt{N}$, where $N$ is the sample size, $\sigma$ is the standard deviation of the measurements and $z_\gamma$ is the quantile set to 1.96 which equals to a 95% confidence

interval. Figure 18 shows the margin of error of the streamwise velocity component for each grid point at a downstream distance of $2\,D$ and $5\,D$. Here it can be seen that the margin of error is higher in the wake in comparison to the ambient air due to the simple fact of a higher standard deviation of the streamwise velocity component. The margin of error is around $0.3\frac{m}{s}$ at $2\,D$ and around $0.15\frac{m}{s}$ at $5\,D$. At some locations a high margin of error is visible due to a low amount of measurement data, caused by the filtering process and the Lissajous pattern resulting in a reduced amount of data at $y/D = 0$ and $z/D = 0$. The

margin of error of the energy dissipation rate is shown in Figure 19, indicating a similar trend visible in Figure 18. Here the



margin of error is around $0.6\frac{m^2}{s^3}$ at 2 D and around $0.4\frac{m^2}{s^3}$ at 5 D. The margin of error has a larger magnitude as the width of the spectrum is squared in Equation 5. This also explains the larger scatter in the dissipation rate contours of Figures 7 and 8, compared to the mean velocity contours of Figures 4 and 5. This indicates that the WindScanner is able to capture the averaged flow properties such as the streamwise velocity and the energy dissipation.

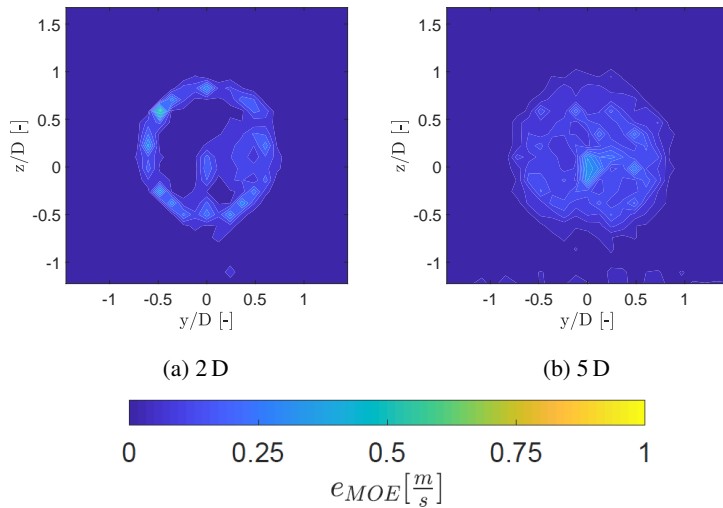

Figure 18. Margin of error ($e_{\mathrm{MOE}}$) of the streamwise velocity at 2 D and 5 D for the inflow without a grid.

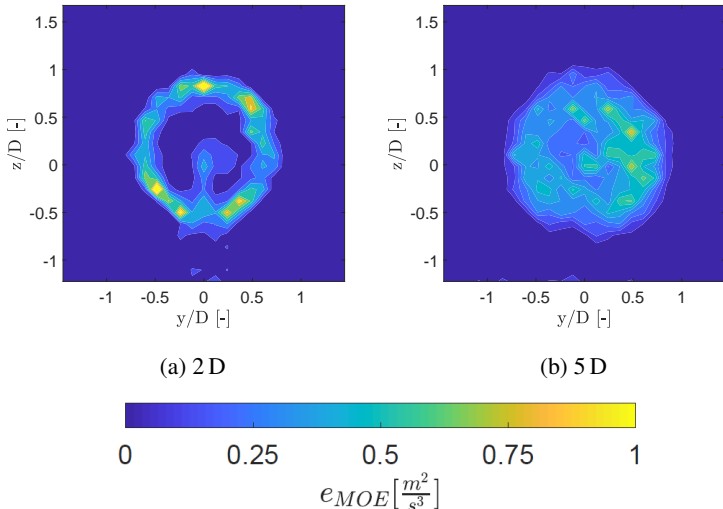

Figure 19. Margin of error ($e_{\mathrm{MOE}}$) of the energy dissipation rate at 2 D and 5 D for the inflow without a grid.





## 5 Conclusions

In general, a strong dependency of the wake characteristics on both the yaw angle and the inflow conditions was observed. Three different inflow conditions were generated, one without a grid, one with a uniform open area passive grid and one with a variable open area passive grid which created an inflow with mean shear approximating a boundary layer. The wakes of a

model wind turbine (MoWiTO 0.6) were measured with turbines operating at maximum power coefficient at a flow speed of $7.5\,m/s$ for rotor yaw angles of $0°$, $+30°$ and $-30°$, and the empty test section flow was measured as well.

Measurements of the free-stream flow within the test section with the noninvasive short-range Lidar WindScanner indicated a stable flow for each inflow condition and no influence of the boundary layer due to the wind tunnel wall on the wake. For the cases with yaw $\psi = \pm 30°$ the wake was deflected laterally up to $0.6\,D$ and a curled wake was observed. The curled wake

develops sooner and stronger in the case of the boundary layer inflow compared to the uniform inflow, suggesting that boundary layer inflow enhances the counter-rotating vortex pair (CVP). The wake deficit distribution in these cases is asymmetric due to the rotation of the turbine and the yaw angle changing the relative wind speed across the rotor plane. The tower wake was observed to be displaced in the opposite direction of the deflection of the turbine wake. The presence of a stronger CVP leads to an asymmetry in the wake deflection with a boundary layer in contrast to the cases with a uniform inflow condition.

The analyses of the energy dissipation rate showed a higher energy dissipation rate within the wake in comparison to the ambient flow. A lower magnitude of the energy dissipation rate is identified at a lower turbulence. In addition, a ring with increased energy dissipation slightly larger than the rotor area and growing in width further downstream is visible for each inflow condition which grows in width further downstream. At $\pm 30°$ yaw the circular shape is stretched to an elliptical shape or a curled shaped, depending on the inflow condition and the downstream distance.

The WindScanner measurements showed a good comparison with the hot-wire data obtained by Neunaber (2019) and the current campaign. However, the WindScanner partially filters out the turbulence due to the Lorentzian spatial weighting function of the measurement device. A similar trend of the temporal averaged streamwise velocity component and the energy dissipation is observed between each data set. However, differences in the magnitude of the temporal averaged streamwise velocity component and the energy dissipation were observed which is related to the filtering process, the sampling frequency,

the probe volume and the method used to determine the energy dissipation from the hot-wire data and the WindScanner. Additionally, an uncertainty analysis showed a relative error of the measurement data up to 3.5% and a margin of error of around $0.3\,\frac{m}{s}$ at $2\,D$ and $0.15\,\frac{m}{s}$ at $5\,D$ for the streamwise velocity component. Furthermore, the measurements of the energy dissipation showed a margin of error around $0.6\,\frac{m^2}{s^3}$ at $2\,D$ and $0.4\,\frac{m^2}{s^3}$ at $5\,D$.

Due to the possibility of mapping the wake fast at multiple locations with the Windscanner, a thorough understanding of the development of the wake is acquired at different inflow conditions and operational conditions. This will aid the process to further improve existing wake models by accounting for the near wake and the dissipation of the wake. Future steps are to compare the acquired data with numerical simulation with the same inflow condition and existing wake models.





*Code and data availability.* The data presented can be made available by contacting the corresponding author. An online database is currently being prepared

*Author contributions.* PH designed the research, conducted the measurement campaign, performed the data analysis, prepared the figures, and planned and wrote the paper. MW assisted the wind tunnel campaign and the data analysis. MH contributed on the operation of the wind

5 tunnel, the active grid and the model wind turbine. MK supervised the work. All coauthors contributed with several fruitful discussions and thoroughly reviewed the manuscript.

*Competing interests.* The authors declare that they have no conflict of interest.

*Acknowledgements.* This work is partly funded by the Federal Ministry for Economic Affairs and Energy according to a resolution by the German Federal Parliament in the scope of research project "CompactWind II" (Ref. Nr. 0325492H) and "DFWind" (Ref. Nr. 0325936C).

10 Martin Wosnik acknowledges support by Hanse-Wissenchaftskolleg (HWK)/Institute of Advanced Study, where he was a Fellow while this work was carried out.





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
