# Peer review of "Development of a Curled Wake of a Yawed Wind Turbine under Turbulent and Sheared Inflow"

_Wind Energy Science, 2021_

## Referee Comment (RC2)

**REVIEW OF WES-2021-65**

*Development of a Curled Wake of a Yawed Wind Turbine under Turbulent and Sheared Inflow*

*authors:*
Paul Hulsman
Martin Wosnik
Vlaho Petrović
Michael Hölling
Martin Kühn

**Summary:**

The manuscript entitled "Development of a Curled Wake of a Yawed Wind Turbine under Turbulent and Sheared Inflow" presents data collected with a continuous wave lidar during a wind tunnel experiment, characterizing the evolution of a yawed wind turbine's wake. The authors state that "the objective is to determine the effect of the boundary layer and turbulence intensity at different yaw angles on the wake deficit, wake deflection and wake dissipation," and is investigated by repeating the experiment in three slightly different inflow cases. The main finding appears to be that "The curled wake develops sooner and stronger in the case of the boundary layer inflow compared to the uniform inflow, suggesting that boundary layer inflow enhances the counter-rotating vortex pair (CVP)," although the connection between the inflow (shear, turbulence, etc.) and the development of the CVP is fairly descriptive. The paper would be greatly strengthened by improving the discussion of the causes of the observed changes, rather than simply describing difference in the flow cases. Additionally, the language in the paper is often convoluted and complex. Simplifying the text will help readers understand the content of the article much more easily.

**Major points:**

- page 1 — The experiment uses a single lidar. How then are dual-Doppler measurements achieved?
- page 2 — What does 'steady and reliable' mean? Steady state? Low uncertainty?
- page 4 — The assumption that $v = w = 0$ must not be true in yawed conditions. Can you quantify with Eq. 2 (to at least the first order) how realistic this assumption is?
- page 4 — What is meant by 'structural resolution'?
- page 5 — Lissajous trajectories for the horizontal and vertical planes appear to have a greater measurement density near the borders of the scan area. Does this lead to a difference in measurement uncertainty? Is error greatest in the center of the scan simply due to the measurements?

- page 8 — What are the min and max wavenumbers considered in equation 8? How are they determined? This description does not provide sufficient detail to understand the methods applied.
- page 8 — "...which remains well below the rotor area (highlighted as a shaded green area)." The boundary layer appears to be nearly up to the rotor area by $13D$ or $16D$, not well clear of it. Is this expected to impact measurements?
- page 9 — It looks like there is some acceleration near $z/D = 0.5$ for $x/D > 0$. Is this from the passive grid?
- page 9 — "air from the outside was sucked into the wind tunnel segment near the bottom leading to the visible speed-up region." This explanation is not clear to me. How would still air from the outside lead to an acceleration near the wind tunnel floor? It seems like air sucked into the tunnel would be slower than the flow in the tunnel.
- page 9, Figure 3 — Does the decay of TI in the uniform passive grid case impact the consistency of results? Is it accounted for in the analysis?
- page 10 — The paragraph discussing turbulence characteristics in the inflows is not clear. For example, in the sentence "This indicates that turbulence is maintained at a near constant level due to the mean shear, which causes production of turbulent kinetic energy." Are the authors saying that that turbulence 'decay' (energy transfer from large scales to small scales, I suppose?) is balanced by production?
- page 11, Figure 4— Interesting that the uniform inflow - no grid has a lower wind speed at the wall than the passive grid case. This seems inconsistent with the description of the inflow cases.
- page 13 — I believe that Martínez et al state that the opposite is true. "We assume that the wake rotation vortex does not decay or deform as it moves downstream. This is not necessarily true, as turbulence mixing will decrease the wake rotation."
- page 14, Figure 6 — I find this figure difficult to follow. Would the evolution of the location of the maximum deficit be more clearly shown with a line plot as a function of downstream distance?
- page 14 — "The turbulent kinetic energy dissipation rate ($\varepsilon$) gives an indication how the flow behaves within the wake: a high dissipation rate indicates a faster mixing of the wake whereas a small dissipation rate suggests that the wake will persist further down stream." I'm not sure that the argument made in this sentence is true. The dissipation rate $\varepsilon$ represents the rate at which turbulent kinetic energy is dissipated into heat, not wake recovery. The key difference here is that wake recovery is typically associated with turbulent mixing and turbulent flux of kinetic energy, not with turbulent dissipation.
- page 13 — "In addition, the difference between the wind speed in the upper part and the lower part reduces for the case with a uniform passive grid shown in Figures 4d to 4f." This sentence is difficult to understand. It seems like the authors are stating that the distribution of momentum deficit in the wake is different than for the no-grid case. Please rephrase.
- page 14 — It is not clear what is meant by "enhanced" dissipation or turbulence. "Enhance" typically means "to make better". Do the authors mean to say "increased"?
- page 14 — The authors frequently refer to the wake of a yawed turbine as being "elliptical," although it does not appear to be. It may be safer to simply state that the wake is no longer axisymmetric due to the CVP or that the curled shape becomes more pronounced moving downstream, as the CVP has had more time/space to deform the $0.9u_\infty$ boundary.
- page 17 —"This also caused that the increment of the energy dissipation rate at hub height is not visible at a downstream distance of $5D$." This sentence is not clear. Are you saying that the turbulent mixing has decreased the dissipation rate following the nacelle?
- page 17 — "...a low dissipation rate is also visible at the region with the highest wake deficit" Turbulent dissipation is related to the gradient of Reynolds stresses. In the center of the wake, those gradients are quite small, so not surprising.

- page 18, Figure 9 — I find the markers on each of the contour lines difficult to distinguish. Would the contours be clearer without the markers? I think the color gradient would be sufficient to distinguish between the lines. Also the caption may be clearer by stating that each line shows the boundary of 0.9 $u_infty$ at each downstream location.
- page 19, Figure 11 — Dotted lines are difficult to see. Please update line styles. Also, the use of color here does not help to distinguish between the different yaw angles or between the spread of wake center locations. Coloring each inflow case does not help interpret the results.
- page 20, Table 2 — Are these values spatially averaged as well? It is not clear why the data in the table are included in the paper, or what they are intended to communicate.
- page 22 — How is the dissipation rate calculated for the hotwire measurements? It might be nice to see how this was calculated to help understand the comparison.

**Minor points:**

- Forward references are used throughout the text. Please try to keep figures near their discussion in the manuscript, and the definition of variables near their appearance in the equations.
- page 8 — The sentence including, "avoiding that the turbulence and wind shear characteristics break down within the measurement domain." is not clear. Please rephrase.
- page 8 — In some cases, descriptions of lengths or distances are in units of meters, while figures describe the domain in terms of rotor diameters. Please be consistent throughout the manuscript.
- page 9 — "turbulence characteristics is crucial" should be "turbulence characteristics are crucial"
- page 10 — As a consideration for the writing style, phrases like "Here it can be detected that, ..." and "Furthermore, ..." do not contribute to the readers understanding of the content of the article and can be safely removed—everywhere.
- page 13 — "transferred to a certain direction" do the authors mean "advected" or "deflected"?
- page 13 — "The transfer of the wake deficit is visualized..." This is not clear. See above.
- page 13 — "detecetion" is repeated on line 29.
- page 22 — "...area which increases at 5D in width." This phrasing makes it sound like the high-dissipation ring is $5D$ wide. Please rephrase.
- page 22 — "can be related due" would be clearer as "attributed"

---

## Author Response (AR1)

**Development of a Curled Wake of a Yawed Wind Turbine under Turbulent and Sheared Inflow**

Paul Hulsman[1], Martin Wosnik[2], Vlaho Petrović[1], Michael Hölling[1], and Martin Kühn[1]

1 - ForWind – Institute of Physics, University of Oldenburg, Küpkersweg 70, 26129 Oldenburg, Germany

2 - University of New Hampshire, Department of Mechanical Engineering, S102 Chase Ocean Engineering Laboratory, 24Colovos Road, Durham NH 03824, United States

**Correspondence**: Paul Hulsman, paul.hulsman@forwind.de

We thank the reviewer for the valuable comments, questions and suggestions. They helped to significantly improve the manuscript. Below are our responses to each comment.

The author's response is shown in red and the rephrased sentences in blue.

**Author response to reviewer 1**

**General comments**

In this research article a comprehensive measurement campaign of the wake behind a yawed wind turbine under different turbulent and sheared inflow conditions is presented. A short-range scanning Lidar system is used to map the wake flow at number of downstream locations, results showing good agreement with reference hot-wire measurements on the same setup. The high quality of Lidar measurements on these small scales is impressive to see, while pre-programmed scanning patterns show great future potential for acquiring full-field wake scans in a wind tunnel in a relative short time.

The paper is well-organized and has a very high quality of language and presentation of data. The results of the time-averaged wake flow for three inflow conditions and yaw angles are supplemented by an analysis of the dissipation rate, which is considered valuable for comparison with previous hot-wire measurements and future CFD simulations of the setup. The main findings on the inflow's influence on wake of yawed turbine confirm findings from previous experiments and simulations, which are cited at the right locations. Given the large amount of research on yawed turbine wakes during the last years, the main findings in this article are not completely novel. The very detailed analysis, well-organized presentation and good discussion of the results contribute to the high quality of this research.

Regarding the content of the article, I only have very few minor comments and technical corrections, which are listed below. Overall, this is an impressive piece of experimental research, and I am looking forward to seeing more following.

**Specific comments:**

P3.L25. Please provide more information on the operational point(s) of the model turbine for the different yaw angles, i.e., thrust and power, if possible. So far, the only info given here is the tip speed ration of 5.7. These would be valuable data for potential CFD simulations or repetitions of the setup.

While the performance ($c_P$, $c_T$) of the specific turbine used has been documented in previous work (e.g., Petrovic et al. 2018), during the measurement campaign reported here only power measurements but no thrust measurements were carried out. The following lines have been added regarding the operational points of the turbine:

*The wind turbine controller is based on the torque of the generator (Petrovic et al. (2018)) leading to a tip speed ratio (TSR) of 5.7 and a power coefficient of $c_P$=0.41 at its operating point for zero-yaw without a grid. For yaw misalignment of +- 30°, the tip speed ratio (TSR) was reduced to 5.3 and the*

*power coefficient was reduced to $c_P=0.29$. The power was measured at the generator, which includes mechanical and electrical losses. During the campaign reported turbine thrust was not measured. However, Neunaber (2019) measured the thrust for this turbine using a force balance and obtained a combined value of the thrust coefficient of the rotor and the drag coefficient of the tower, related to the swept area, equal to $c_{T_{Rotor}} + c_{D_{Tower}}^* = 1.0$. This value was also measured without a grid and for zero-yaw.*

P10.L21. lateral displacement of the tower wake in yawed conditions. "This is a result of conservation of mass…". How much is the streamwise distance of from the rotor center to the tower center? Could this contribute to a lateral displacement of the tower wake when the turbine is yawed?

The distance between the rotor plane and the tower is 110mm for the MoWiTO 0.6. During a yaw-misalignment of 30 the rotor centre is shifted by 55mm. In Figure 5, the tower wake is displaced to the opposite than the deflection of the wake. The figures are all centred at the tower centre, meaning that the streamwise distance of the rotor centre to the tower centre does not result into a lateral displacement. The description in the methodology section has been modified to clarify how the scans were performed.

*P3 The distance between the rotor centre and the tower centre is 110mm (0.19D).*

*P5 Furthermore, the dimension of the vertical plane is shown in Figure 1c with an area equalling to approximately 3D x 3D (1.74m x 1.74m) with y = 0[m] indicating the wake centre at non-misaligned cases and the tower centre*

P14.L13. "The high dissipation rate at the wake centre can be related to the root vortices within the near wake." I do not really see any significant signature of root vortices in the plots describing the near wake at 2D. There seems to be one "yellow dot" in the central wake in Fig.7(e), but is this really a root vortex signature? Why isn't it visible in Fig. 7(b)?

The increase in the dissipation rate at the wake centre is visible in 7(b) and 7(e) in the blue color shading. At the centre the blue shading is lighter than at a larger radius (darker blue shading). The increase of the energy dissipation is not as profound as at the wake centre. The sentenced has been changed to indicate that a slight increase in the energy dissipation is visible.

*The slightly higher dissipation rate at the wake centre visible in Figure 7b and Figure 7e in the different colour shading at the centre can be related to the root vortices within the near wake.*

P28.L1. "An online database is currently being prepared." That is a very good idea to make this extensive dataset available for validation purposes. I hope the authors can provide a link to the dataset in the final version of this paper.

An online database has been prepared. At the moment we are working on finding a platform to share the data.

**Technical corrections:**

P4.L8. "Equations 2 approximates…" -> Equation

Has been changed

P9.L11. "… the turbulence is higher initially and the rapidly decays moving downstream …" -> it

Has been changed

P20.L17. "… which can casue spatial averaging …" -> cause

Has been changed

P21.L4. "At 1D in Figure 12…" -> Figure 13?

Has been changed

**Author response to reviewer 2**

The manuscript entitled "Development of a Curled Wake of a Yawed Wind Turbine under Turbulent and Sheared Inflow" presents data collected with a continuous wave lidar during a wind tunnel experiment, characterizing the evolution of a yawed wind turbine's wake. The authors state that "the objective is to determine the effect of the boundary layer and turbulence intensity at different yaw angles on the wake deficit, wake deflection and wake dissipation," and is investigated by repeating the experiment in three slightly different inflow cases. The main finding appears to be that "The curled wake develops sooner and stronger in the case of the boundary layer inflow compared to the uniform inflow, suggesting that boundary layer inflow enhances the counter-rotating vortex pair (CVP)," although the connection between the inflow (shear, turbulence, etc.) and the development of the CVP is fairly descriptive. The paper would be greatly strengthened by improving the discussion of the causes of the observed changes, rather than simply describing difference in the flow cases. Additionally, the language in the paper is often convoluted and complex. Simplifying the text will help readers understand the content of the article much more easily.

We have gone over the entire text and changed the sentences that were convoluted and complex. The text is simplified and is written more descriptive to aid the reader. The discussion has been further improved to describe the causes of the measured changes.

Major points:

•page 1 — The experiment uses a single lidar. How then are dual-Doppler measurements achieved?

Dual-Doppler measurements were not achieved. The sentence was altered.

*A short-range Lidar WindScanner facilitated mapping the wake with a high spatial and temporal resolution in vertical, cross-stream planes at different downstream locations and in a horizontal plane at hub height.*

•page 2 — What does 'steady and reliable' mean? Steady state? Low uncertainty?

A steady and reliable yaw-control model indicates that it can accurately predict the required yaw-misalignment at a given inflow condition to increase the power gain. This sentence has been rephrased.

*Experimental data under controlled conditions are necessary to understand the wake behaviour at different yaw angles and at different inflow conditions which is critical for developing a reliable and accurate yaw control model which can be implemented in the field.*

•page 4 — The assumption that v = w = 0 must not be true in yawed conditions. Can you quantify with Eq. 2 (to at least the first order) how realistic this assumption is?

The Section 'Uncertainty Analysis' quantifies how realistic this assumption is during non-misaligned cases. Here an uncertainty of 1m/s for the w- and v-component at 2D resulted in a maximum uncertainty of 3.1% for the u-component. As in the near wake during yawed cases a large w- and v-component is expected, which results to a similar uncertainty. These cases are not shown in the paper to reduce the content and avoid confusion for the reader. In the Section 'Uncertainty Analysis' a sentence has been added to indicate how the uncertainty behaves during yaw-misalignment.

*P4 Within the near-wake region a large lateral and vertical velocity component is expected, which influences the calculation for the streamwise velocity component. The uncertainty is further discussed in Section 4.2. Further downstream the lateral and vertical velocity component reduces, and the assumption fits better.*

•page 4 — What is meant by 'structural resolution'?

'Structural resolution' refers to the resolution of the flow structures. The sentence has been changed to 'spatial resolution'

*It allows measurements of airflow velocity at reasonably high temporal and spatial resolution without disturbing the flow.*

•page 5 — Lissajous trajectories for the horizontal and vertical planes appear to have a greater measurement density near the borders of the scan area. Does this lead to a difference in measurement uncertainty? Is error greatest in the center of the scan simply due to the measurements?

It is correct that the difference in measurement density leads to a difference in measurement uncertainty. The difference in the measurement uncertainty is shown in Section 4.2 'Uncertainty Analysis', where the margin of error within a scan area is shown. Figure 18-19 indicates that the largest influence on the margin of error is the standard deviation of the flow. If the standard deviation is similar in the entire scan area than a larger error is expected in the scan centre. A sentence has been added in this section referring to Section 4.2.

The effect due to the measurement density on the margin of error is discussed in Section 4.2.

•page 8 — What are the min and max wavenumbers considered in equation 8? How are they determined? This description does not provide sufficient detail to understand the methods applied.

The value for kmax can be found by identifying the local minimum from the energy spectrum. The value for kmin is determined by identifying the wavelength at which the energy spectrum corresponds to $E(f) \propto f^{-5/3}$. The paragraph has been modified accordingly.

*The value for kmax can be found by identifying the local minimum from the energy spectrum. The value for kmin is determined by identifying the wavelength at which the energy spectrum corresponds to $E(f) \propto f^{-5/3}$*

•page 8 — "...which remains well below the rotor area (highlighted as a shaded green area)." The boundary layer appears to be nearly up to the rotor area by 13D or 16D, not well clear of it. Is this expected to impact measurements?

Based on the results of the mean wake flow at 13D and 16D it is concluded that the boundary layer does not heavily influence the wake shape and does not significantly influence the results, see Figure A.1. However, it is expected that the boundary layer effects the wake more in the latter two cases in comparison to the other cases shown in the paper.

[Figure]

Figure A.1: Mean flow at laminar inflow condition at 13D (Left) and 16D (right)

The following line has been added:

*Analysis of the mean wake flow at 13D and 16D indicated a minor influence of the boundary layer on the wake shape.*

•page 9 — It looks like there is some acceleration near z/D = 0.5 for x/D > 0. Is this from the passive grid?

A minor speed up equal to 1.3% of the mean inflow with a passive grid is indeed visible between x = 0D and x = 1D. This could be related to the decay of the turbulent structures, visible in Figure 3. It is assumed that the small speed up has a minimal effect on the wake behaviour as the inflow remained stable between 1D up to 10D. The paragraph describing the inflow condition with the passive grid has been modified.

*A small speed-up of 1.3% between 0D and 1D is visible, due to the decay of the turbulent structures. It is assumed that this has a minor effect on the wake behaviour as the inflow condition remained stable with no speed-up between 1D up to 10D.*

•page 9 — "air from the outside was sucked into the wind tunnel segment near the bottom leading to the visible speed-up region." This explanation is not clear to me. How would still air from the outside lead to an acceleration near the wind tunnel floor? It seems like air sucked into the tunnel would be slower than the flow in the tunnel.

During further investigation we have noticed that the speed-up region near the bottom is due to the position of the first flap of the active grid. To recreate a sheared inflow, the active grid changes the blockage using the flaps over the entire inlet-area. However, the area between the floor of the wind tunnel segment and the first flap is smaller in comparison to the area between the first and second row of the active grid. This leads to the visible speed-up region near the bottom of the wind tunnel segment. This section has been rewritten.

*A possible cause of the speed-up is the distribution of the blockage over the inlet area. Due to the positioning of the first flap, the blockage between the wind tunnel segment floor and the first row of flaps is lower in comparison to the blockage between the first and second row of flaps.*

•page 9, Figure 3 — Does the decay of TI in the uniform passive grid case impact the consistency of results? Is it accounted for in the analysis?

It is assumed that the decay of TI does not impact the consistency of the results as the reduction is from 1.5% to 0.7%. Furthermore, the mean inflow condition showed a very constant flow. This is mentioned in Section 3.1.

•page 10 — The paragraph discussing turbulence characteristics in the inflows is not clear. For example, in the sentence "This indicates that turbulence is maintained at a near constant level due to the mean shear, which causes production of turbulent kinetic energy." Are the authors saying that that turbulence 'decay' (energy transfer from large scales to small scales, I suppose?) is balanced by production?

Yes, some of the statements discussing the different turbulent inflow cases were not clear. We were trying to highlight the difference between the passive uniform grid and the variable open area grid. The variable area open grid appears to reach a stable level of turbulence much quicker, which is attributed to the mean shear and turbulent kinetic energy production.

We changed the sentences to clarify this part:

*This suggests that turbulence is maintained at a near constant level due to the mean shear, which causes production of turbulent kinetic energy between $1 \leq x/D \leq 10$. The variable open area reduces the generated turbulence length scale at the grid and thus leads to a faster decay of the turbulence intensity in the region between the grid and $1x/D$*

We also noted that $x/D=0$ in the turbine coordinate system used in Figure 3 is at 2.4 D downstream of the grid.

•page 11, Figure 4— Interesting that the uniform inflow - no grid has a lower wind speed at the wall than the passive grid case. This seems inconsistent with the description of the inflow cases.

In Figure 2 a lower wind speed at the wall is also visible for the uniform – no grid inflow case, which is consistent with the results shown in Figure 4.

•page 13 — I believe that Martínez et al state that the opposite is true. "We assume that the wake rotation vortex does not decay or deform as it moves downstream. This is not necessarily true, as turbulence mixing will decrease the wake rotation."

'The increase of the rotation strength further downstream' was related to the strength of the vortices during yaw-misalignment. This sentence has been rewritten as it was not clearly formulated.

*The region with the highest wake deficit is pushed to a certain direction depending on the yaw angle, highlighted in Figure 6. The movement is due to the combination of the wake rotation and the CVP described in Martínez-Tossas et al. (2019)*

•page 14, Figure 6 — I find this figure difficult to follow. Would the evolution of the location of the maximum deficit be more clearly shown with a line plot as a function of downstream distance?

A line plot would only be able to show the evolution of the maximum deficit in one axis. The purpose of the Figure is to show that the deficit is pushed in a certain direction depending on the yaw angle. In addition, by representing the data as shown in Figure 6, the circular motion of the wake deficit is better visible.

•page 14 — "The turbulent kinetic energy dissipation rate ($\varepsilon$) gives an indication how the flow behaves within the wake: a high dissipation rate indicates a faster mixing of the wake whereas a small dissipation rate suggests that the wake will persist further down stream." I'm not sure that the argument made in this sentence is true. The dissipation rate $\varepsilon$ represents the rate at which turbulent kinetic energy is dissipated into heat, not wake recovery. The key difference here is that wake recovery is typically associated with turbulent mixing and turbulent flux of kinetic energy, not with turbulent dissipation.

We agree that the wake recovery is associated with turbulent mixing. The turbulent kinetic energy dissipation rate is typically increased for higher turbulent mixing within a wake, and thus the dissipation rate can be seen as an indirect indicator of how quickly the flow will recover. The higher dissipations rate measurements indicate faster wake recovery. The difference in turbulent kinetic energy, estimated from only the streamwise component, shows trends comparable to the dissipation rate. We changed the text to clarify this and reflect the reviewer comments.

*The turbulent kinetic energy dissipation rate ($\varepsilon$) is typically increased for increased turbulent mixing within a wake. Increased turbulent mixing leads to faster wake recovery and thus, the dissipation rate can be seen as an indirect indicator of how the flow will evolve within the wake: a high dissipation rate suggests more mixing of the wake whereas a small dissipation rate suggests less mixing and that the wake will persist further downstream.*

•page 13 — "In addition, the difference between the wind speed in the upper part and the lower part reduces for the case with a uniform passive grid shown in Figures 4d to 4f." This sentence is difficult to understand. It seems like the authors are stating that the distribution of momentum deficit in the wake is different than for the no-grid case. Please rephrase.

This sentence has been rephrased:

*In addition, the difference between the wind speed deficit in the upper part and the lower part seems to be reduced for the case with a uniform passive grid shown in Figures 4d to 4f. Both cases were conducted with a uniform inflow condition and the same wind speed, which indicates that the distribution of the momentum deficit behind the rotor should be equal. This suggests that the difference is related to the higher turbulence intensity and the higher mixing rate.*

•page 14 — It is not clear what is meant by "enhanced" dissipation or turbulence. "Enhance" typically means "to make better". Do the authors mean to say "increased"?

This has been changed to 'increased.

•page 14 — The authors frequently refer to the wake of a yawed turbine as being "elliptical," although it does not appear to be. It may be safer to simply state that the wake is no longer axisymmetric due to the CVP or that the curled shape becomes more pronounced moving down-stream, as the CVP has had more time/space to deform the $0.9u\infty$ boundary.

The usage of the word 'ellipitical' has been omitted when the wake shape is described.

•page 17 —"This also caused that the increment of the energy dissipation rate at hub height is not visible at a downstream distance of 5D." This sentence is not clear. Are you saying that the turbulent mixing has decreased the dissipation rate following the nacelle?

Yes, we are highlighting that at 5D the dissipation rate has decreased due to turbulent mixing. The sentence has been reformulated.

*Due to the breakdown of the root vortices in the near wake, the increment of the energy dissipation rate at hub height is not visible at a downstream distance of 5D.*

•page 17 — "...a low dissipation rate is also visible at the region with the highest wake deficit" Turbulent dissipation is related to the gradient of Reynolds stresses. In the center of the wake, those gradients are quite small, so not surprising.

Your comment related to the Reynolds stresses has been added:

*In addition, similar to the flow at 2D a low dissipation rate is also visible at the region with the highest wake deficit, as it is related to the gradient of Reynolds stresses.*

•page 18, Figure 9 — I find the markers on each of the contour lines difficult to distinguish. Would the contours be clearer without the markers? I think the color gradient would be sufficient to distinguish between the lines. Also the caption may be clearer by stating that each line shows the boundary of 0.9 uinfty at each downstream location.

We have tried creating this plot only with the color gradient. However, the lines become very difficult to distinguish. We found that using the markers made it easier to follow the lines. We have recreated the Figure with fewer markers for each line, see Figure A.2.

[Figure]

Figure A.2: Growth of the curled wake behind the wind turbine model at multiple downstream distances. Left: Olde version. Right: New version

The caption has also been modified:

*Growth of the curled wake behind the wind turbine model at multiple downstream distances. The contour-lines indicate the boundary with $\frac{u_{i,\infty} - u_i}{u_{i,\infty}} = 0.9$ at each downstream location. The view is looking upstream towards the turbine model.*

•page 19, Figure 11 — Dotted lines are difficult to see. Please update line styles. Also, the use of color here does not help to distinguish between the different yaw angles or between the spread of wake center locations. Coloring each inflow case does not help interpret the results.

The line styles have been updated and the colouring for each inflow case has been modified.

[Figure]

*Figure 3: Wake deflection at different inflow condition. Left: Old version. Right: New version*

•page 20, Table 2 — Are these values spatially averaged as well? It is not clear why the data in the table are included in the paper, or what they are intended to communicate.

The intention of the table is to show the difference of the temporal averaged flow properties between the WindScanner measurements and the hot-wire measurements. However, for the reader this can be seen as repetative as multiple cases are used to compare the averaged flow data of the WindScanner with hot-wire data. This paragraph has been removed.

•page 22 — How is the dissipation rate calculated for the hotwire measurements? It might be nice to see how this was calculated to help understand the comparison.

The method to calculate the dissipation rate has been added to Section 2.4

Minor points:

•Forward references are used throughout the text. Please try to keep figures near their discussion in the manuscript, and the definition of variables near their appearance in the equations.

We have gone through the paper and corrected the positions of the figures. We did the same for the definitions of the variables.

•page 8 — The sentence including, "avoiding that the turbulence and wind shear characteristics break down within the measurement domain." is not clear. Please rephrase.

This sentence has been rephrased:

*To conduct a meaningful analysis of wake characteristics a well-defined flow field is required. It is preferred that the turbulence and wind shear characteristics do not significantly change within the measurement domain.*

•page 8 — In some cases, descriptions of lengths or distances are in units of meters, while figures describe the domain in terms of rotor diameters. Please be consistent throughout the manuscript.

We have gone over the entire paper and changed the description of the lengths in terms of rotor diameters.

•page 9 — "turbulence characteristics is crucial" should be "turbulence characteristics are crucial"

This has been changed

•page 10 — As a consideration for the writing style, phrases like "Here it can be detected that, ..." and "Furthermore, ..." do not contribute to the readers understanding of the content of the article and can be safely removed—everywhere.

We have gone over the entire paper and rephrased the sentences starting with 'Here it can be ... ' and 'Furthermore,...'

•page 13 — "transferred to a certain direction" do the authors mean "advected" or deflected"?

Has been changed to 'pushed', which is the same formulation used in Martínez-Tossas et al. (2019)

•page 13 — "The transfer of the wake deficit is visualized..." This is not clear. See above.

Has been rephrased to the following:

*The region with the highest wake deficit is pushed to a certain direction depending on the yaw angle, highlighted in Figure 6. The movement is due to the combination of the wake rotation and the CVP described in Martínez-Tossas et al. (2019)*

•page 13 — "detecetion" is repeated on line 29.

Has been changed.

•page 22 — "...area which increases at 5D in width." This phrasing makes it sound like the high-dissipation ring is 5D wide. Please rephrase.

Has been changed to the following:

*The reduction agrees with Figure 7b and Figure 8b, showing a ring with a high energy dissipation rate slightly larger than the rotor area which increases in width at downstream location of 5D.*

•page 22 — "can be related due" would be clearer as "attributed"

Has been changed.